# Enhancing Deception Detection with Cognitive Load Features: An Audio-Visual Approach

## Abstract

Deception ranges from minor mischief to serious fraud, often leading to significant psychological and financial harm. Effective deception detection is crucial to mitigate these risks and preserve societal trust. Cognitive load is a useful indicator for detecting deception, as lying causes individuals to experience greater mental strain. While prior research leveraged cognitive load features, typically measured through physiological signals such as pupil dilation, these methods often require specialized equipment and can be subject to human bias. These limitations hinder the scalability and automation of deception detection systems. Thus, we propose a novel deception detection framework that automatically extracts cognitive load features from audio-visual data, eliminating the need for specialized hardware or subjective human input. Our approach integrates these features into the deception detection pipeline, enhancing its robustness. Moreover, we introduce a focal loss to address the inherent complexity of deception detection. This objective function enables the model to focus on harder-to-detect instances of deception, thereby improving the performance. Our approach achieves state-of-the-art results on benchmark audio-visual datasets, demonstrating improvements in automated deception detection. Extensive experiments validate the effectiveness of both our cognitive load feature extraction and the proposed objective function in advancing the field.

## 1 Introduction

Deception, the act of leading someone to believe false information as true, has been a subject of ethical debate since ancient times. For millennia, lying has been viewed as a moral issue–St. Augustine regarded every lie as a sin, while philosophers such as Aristotle and Kant expressed similarly strong stances against deception (Zuckerman, 1981). In contemporary society, deception poses significant challenges. It affects critical areas such as judicial proceedings, security protocols, and public trust, with far-reaching consequences, including fraud and other societal harm. As deception is pervasive in everyday life (Wu et al., 2018) and often serves as a social tool (Guerrero et al., 2017), there is a growing demand for reliable, automated methods of deception detection across domains such as airport security, criminal investigations, job interviews, and marketing (Pérez-Rosas et al., 2015b).

However, the human ability to detect deception is limited. Individuals overestimate their ability to identify deception in others while underestimating their capacity to deceive (Elaad, 2003). Empirical studies suggest that the average person detects deception with approximately 54% accuracy—barely better than chance (Bond Jr & DePaulo, 2006). This limitation leads to substantial research to develop more accurate and efficient deception detection techniques.

Recent deception detection work examined verbal and non-verbal cues to differentiate between truth-tellers and liars. Verbal deception detection strategies include interrogation techniques, the role of communication mode in deception, and structured interview approaches such as the PEACE model (Bull et al., 2019; Hartwig et al., 2011; Strömwall & Anders Granhag, 2003). Non-verbal research tended to focus on behavioral patterns during repeated questioning, the impact of speech disfluency and gestures, and training methods to improve detection accuracy through non-verbal cues (Fiedler & Walka, 1993; Granhag & Strömwall, 2002; King et al., 2020). Most prior research underscored the effectiveness of verbal and non-verbal indicators in identifying deception.

A key reason differentiating liars from truth-tellers is the cognitive load associated with deception. Lying imposes higher cognitive demands than truth-telling, which can manifest in both verbal and non-verbal behaviors (DePaulo et al., 2003b). Prior research explored the link between cognitive load and deception detection, suggesting that liars experience greater cognitive strain, which can be leveraged to improve detection accuracy (Bird et al., 2019; Blandón-Gitlin et al., 2014; Van't Veer et al., 2014; Wielgopolan & Imbir, 2023). To measure cognitive load, previous research examined physiological signals such as pupil dilation, blink rates, body movements, and response times (Abouelenien et al., 2016; Constâncio et al., 2023; Elkins et al., 2012; Raiman et al., 2011; Walczyk et al., 2012). Despite the promise of these approaches, practical limitations remain—reliance on specialized equipment, inconsistent physiological interpretations, and challenges in real-time analysis restrict their scalability and portability (Joseph, 2013; Vanneste et al., 2021; Weber et al., 2021).

Inspired by the potential of cognitive load for deception detection, we propose a novel approach that incorporates audio-visual cognitive load features into a fully automated deception detection framework. Our proposed framework, **AVDDCL** (Audio-Visual Deception Detection with Cognitive Load), extracts cognitive load features directly from audio-visual data, overcoming the limitations of traditional methods that rely on specialized equipment or manual annotations. By automating the detection process, our method offers a scalable and efficient solution for real-world applications.

To the best of our knowledge, this is one of the beginning steps in utilizing audio-visual cognitive load features for deception detection. Our contributions are presented as follows:

- We introduce a novel framework that integrates audio-visual cognitive load features into deception detection, moving beyond conventional physiological and behavioral analyses. This approach offers a more scalable, automated, and comprehensive solution.

- We introduce the focal loss to address the specific challenges of deception detection, where differentiating between truth and deception is inherently difficult. By focusing on harder-to-detect cases, our model improves overall detection accuracy.

- Our proposed approach establishes a new state-of-the-art performance benchmark on the DOLOS dataset (Guo et al., 2023), demonstrating significant improvements in both accuracy and robustness for fully automated deception detection systems.

## 2 RELATED WORKS

### 2.1 DECEPTION DETECTION APPROACHES

Deception detection is extensively investigated through various approaches, ranging from analyzing physiological patterns to verbal and non-verbal cues. Early deception detection methods mainly relied on physiological indicators such as heart rate, blood pressure, and skin conductivity, employing polygraph-based techniques. Despite their widespread use, polygraph tests faced consistent criticism among scientists due to concerns over their validity (Meijer & Verschuere, 2014).

Following the limitations of physiological methods, the research mainstream shifted toward non-verbal cues, emphasizing facial expressions and behavioral indicators. Ekman's work on deception detection through facial expressions became foundational (Ekman, 2009), while subsequent studies explored the use of micro-expressions (Wu et al., 2018) and body movement analysis (Van der Zee et al., 2019). Another line of research focuses on gaze tracking and eye interaction as deception indicators (Kumar et al., 2021; Mirsadikov & George, 2023). However, micro-expressions could not significantly enhance deception detection accuracy, leading to contradictory conclusions about their effectiveness (Jordan et al., 2019). Several methods often required advanced and non-portable equipment, limiting their practical applications in real-world settings (Dinges et al., 2023).

Recently, advancements in deep learning have propelled detection research beyond single-modality approaches, encouraging multi-modality fusion. Several prior research integrated diverse information sources (channels), including audio, visual, and EEG signals, into deep learning frameworks for deception detection (Şen et al., 2020). Others combined physiological responses, thermal sensing, and linguistic features to achieve multimodal deception detection (Abouelenien et al., 2014).

## 2.2 DECEPTION DETECTION WITH COGNITIVE LOAD

Previous research suggested that liars experience more significant cognitive load and nervousness than truth-tellers, as they exert additional mental effort to appear credible (Vrij, 2008). In many situations, telling the truth is cognitively less demanding than lying, especially in face-to-face interactions, where fabricating a lie requires more mental resources than simply recounting the truth (Van't Veer et al., 2014; Vrij, 2008). Lying imposes a cognitive burden, making it more effortful than truth-telling (Van't Veer et al., 2014), and liars often exhibit detectable signs of increased cognitive load, such as subtle behavioral cues (Blandón-Gitlin et al., 2014). Notably, the effectiveness of deception detection methods is often linked to the degree of cognitive load experienced by the individual, with higher cognitive load improving detection accuracy (Wielgopolan & Imbir, 2023).

Several scholars utilized physiological indicators of cognitive load to improve deception detection. For example, eye blinking and pupil dilation are closely associated with cognitive load (Stern et al., 1984), making them reliable markers for detecting deception through eye-tracking technologies and pupil diameter measurements (Labibah et al., 2018). Other scholars reinforced this finding, demonstrating that increased pupil dilation during deception was a significant indicator of cognitive load (Pasquali et al., 2020). Moreover, eye movement, response time, and eloquence were effective deception indicators, as they were strongly tied to cognitive load (Gonzalez-Billandon et al., 2019). Micro-expressions on the face, particularly under cognitive strain, were also examined to enhance deception detection accuracy, with research suggesting that imposing cognitive demands during interviews could significantly improve the identification of deceptive behaviors (Monaro et al., 2022).

Although physiological methods have the potential for assessing cognitive load in deception detection, their practical application is limited by the need for specialized and non-portable equipment (Weber et al., 2021). Furthermore, individual variability in physiological responses introduces challenges in data interpretation (Joseph, 2013), and the complexity of real-time assessment complicates practical deployment (Vanneste et al., 2021).

To overcome these limitations, the automatic extraction of cognitive load features presents a promising alternative. For example, the AVCAffe dataset (Sarkar et al., 2023) employs audio-visual data to analyze cognitive load and affective states. Building on this foundation, we introduce the first automated method to extract cognitive load indicators from audio-visual data specifically for deception detection, filling a significant gap in the current literature. Considering cognitive load features as intermediate tasks, we propose a novel multimodal framework that improves deception detection performance while overcoming the constraints related to traditional physiological methods. It provides us a foundation for scalable, practical applications in real-world deception detection scenarios.

## 3 PROPOSED METHOD

We propose a novel framework that integrates cognitive load features for deception detection. The framework comprises a feature extraction network and a deception detection network. Figure 1 shows the overview of the architecture of AVDDCL.

### 3.1 FEATURE EXTRACTION NETWORK

We extract audio-visual features from the audio-visual parameter efficient fusion (AVPEF). We use the feature extraction network proposed by the prior study (Guo et al., 2023) for efficient feature extraction. Figure 2(a) shows the overall structure of AVPEF. AVPEF consists of a uniform temporal encoder (UTE) and audio-visual fusion (AVF). In AVPEF, the input audio and visual data are tokenized separately through 1D-CNN and 2D-CNN modules, and input data is converted into sequential representations. These sequences then pass UTE, which is based on W2V2 (Baevski et al., 2020) and ViT (Dosovitskiy et al., 2021)-based transformer encoders, to extract modality-specific features. The audio-visual integrated feature combines audio and visual features through AVF.

**Uniform Temporal Encoder**. The uniform temporal encoder (UTE) blocks are stacked in the AVPEF to focus on temporal information. $\text{UTE}_A$ consists of the pre-trained W2V2 encoder, and $\text{UTE}_V$ consists of the pre-trained ViT encoder. Given that W2V2 and ViT are large pre-trained models, fully fine-tuning them can be inefficient and may lead to overfitting. So we only fine-tune a small number of additional parameters with a uniform temporal adapter (UT-Adapter). With UTE,

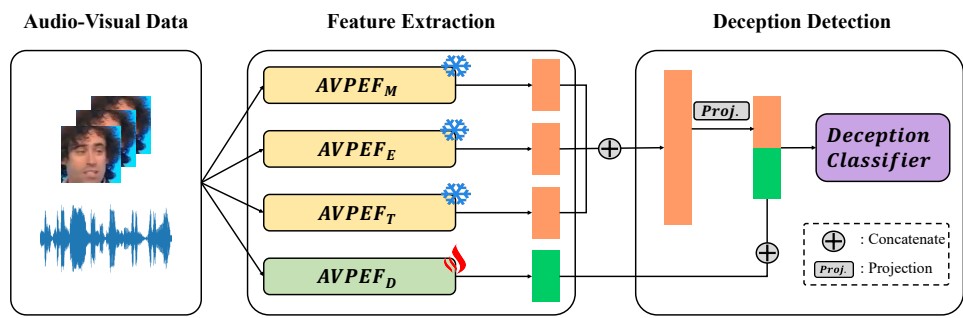

Figure 1: Overview of AVDDCL. AVDDCL receives audio-visual data as input and extracts audio-visual features through AVPEF (Sec. 3.1). The cognitive load features are extracted from the pre-trained AVPEF network (Sec. 3.2) and are concatenated with deception features. The deception classifier (Sec. 3.3) detects the deception with the concatenate features.

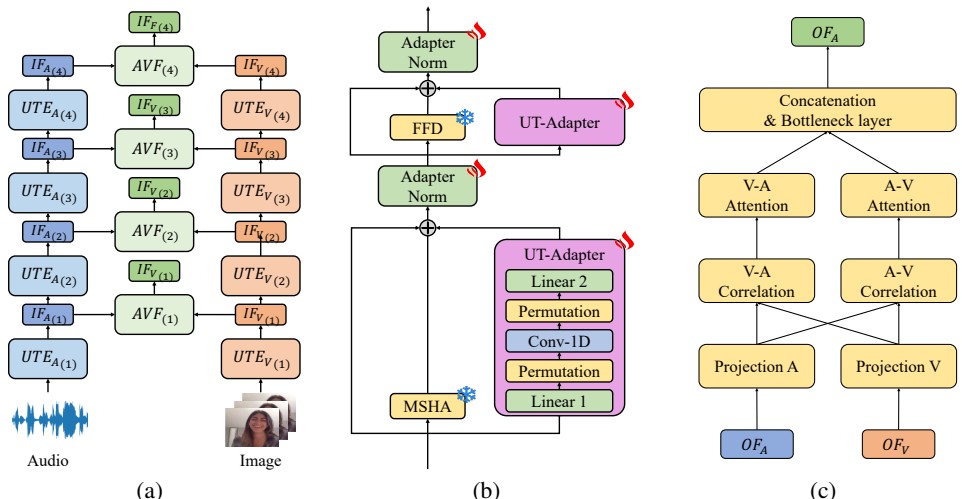

Figure 2: Overview of the Audio-Visual Parameter-Efficient Fusion network (AVPEF); (a) The overall AVPEF structure, (b) Uniform Temporal Encoder (UTE), (c) Audio-Visual Fusion (AVF).

our proposed model improves the parameter efficiency and avoids the risk of overfitting. The structure of UTE is depicted in Figure 2(b). The UT-Adapter is designed to capture local temporal dynamics, complementing the Multi-Head Self-Attention (MSHA) module, which primarily captures global temporal and spatial attention.

**Audio-Visual Fusion**. The audio-visual fusion (AVF) module facilitates the efficient fusion of audio and visual features. The intermediate features of audio ($IF_A$) and visual ($IF_V$) encoders are projected into a lower-dimensional embedding space to reduce computational costs. After the projection, the cross-modal correlations are calculated using a trainable weight matrix. Then cross-modal attention is applied to refine the feature representations. The refined audio and visual features are concatenated and the fused feature ($F_{AV}$) is obtained through the bottleneck layer.

**Audio-Visual Feature Extraction**. As the sequence of input audio ($X_A$) and image ($X_V$) are input in AVPEF, each intermediate feature $IF_A$ and $IF_V$ are extracted from the UTE block. The audio-visual fusion feature ($F_{AV}$) is calculated through AVF, and this process is repeated. From the second UTE block, the output of the previous block is received as input. In this study, four UTE blocks are utilized, and the resulting fused representations $F_{AV(1)}, F_{AV(2)}, F_{AV(3)}, F_{AV(4)}$ as follows:

$$IF_{(i)} = \begin{cases} UTE_{(i)}(X) & \text{if } y = 1 \\ UTE_{(i)}(IF_{(i-1)}) & \text{otherwise.} \end{cases} \tag{1}$$

$$F_{AV_{(i)}} = AVF_{(i)}(IF_{A_{(i)}}, IF_{V_{(i)}}) \tag{2}$$

The fused features are concatenated and form the final audio-visual features $F$ as defined below. The final feature $F$ is subsequently used as the input for the classification layer to detect deception.

$$F = F_{AV_{(1)}} \oplus F_{AV_{(2)}} \oplus F_{AV_{(3)}} \oplus F_{AV_{(4)}} \tag{3}$$

## 3.2 COGNITIVE LOAD FEATURE PRE-TRAINING

To extract cognitive load features, we employ the pre-trained AVPEF network, specifically designed to capture the critical role of temporal information in cognitive load assessment (Liu et al., 2023; Li et al., 2025; Puma et al., 2018). The network's UT-Adapter effectively incorporates temporal dynamics, while its parameter efficiency ensures it remains computationally feasible, even in resource-constrained environments. For pre-training, we utilize the AVCAffe dataset (Sarkar et al., 2023), which includes 58,112 short video segments averaging 6.74 seconds each, totaling approximately 108.72 hours of data from 106 participants. These segments were extracted from longer task-based recordings using a silence detection algorithm (Robert et al., 2018) to capture affective and cognitive load attributes with high temporal granularity. This approach enables a more precise and scalable analysis of cognitive load features, which is essential for advancing deception detection.

The dataset provides task-based self-reported labels for arousal, valence, and cognitive load attributes. Cognitive scores, based on NASA-TLX (Hart, 2006), are rated on a 0-21 scale across categories such as mental demand, physical demand, temporal demand, performance, effort, and frustration. Scores above 10 are classified as high, and others as low. In the AVCAffe dataset, frustration, physical demand, and performance were excluded due to minimal variance, with the focus placed instead on mental demand, effort, and temporal demand.

### 3.2.1 PRE-TRAINED AVPEF

We pre-trained the AVPEF network to extract cognitive load features across three key dimensions: mental demand, effort, and temporal demand. During pre-training, we integrated linear layers to predict each of these dimensions from the audio-visual dataset, training them separately to capture specific aspects of cognitive load. To ensure robust model evaluation and prevent information leakage, we divided the dataset into 86 participants for training and 20 for validation, carefully stratified by age, gender, and ethnicity, with no overlap in recording sessions. The AVPEF module effectively learns distinct audio-visual patterns associated with each cognitive load dimension. For the final deception detection task, we removed the linear layers, retaining only the pre-trained AVPEF network for feature extraction, ensuring a streamlined and effective analysis pipeline.

## 3.3 AUDIO-VISUAL BASED DECEPTION DETECTION

### 3.3.1 DETECTION PROCEDURES

As the audio and visual data are input, audio-visual features are extracted from the feature extraction network. We extract four audio-visual features, three for cognitive load features, and one for deception features. For cognitive load features, we utilize the pre-trained AVPEF and freeze its weights. This ensures that the network parameters remain fixed during feature extraction. This process is represented as follows:

$$F_{\mathrm{M}} = \mathrm{FrAVPEF_M}(X), \quad F_{\mathrm{E}} = \mathrm{FrAVPEF_E}(X), \quad F_{\mathrm{T}} = \mathrm{FrAVPEF_T}(X) \tag{4}$$

where FrAVPEF denote the Frozen AVPEF module, $F_{\mathrm{M}}$, $F_{\mathrm{E}}$, and $F_{\mathrm{T}}$ denote the extracted cognitive load features for the mental demand, effort, and temporal demand categories, respectively. X represents the input data. The output of the Frozen AVPEF module for cognitive load is collectively referred to as $F_C$. $F_C$ comprehensively represents the participant's cognitive state. By combining these three dimensions, the model captures a holistic view of the cognitive load experienced by individuals, which is then utilized for subsequent prediction and analysis.

Meanwhile, for deception features ($F_D$), the AVPEF module remains fully learnable, allowing it to capture the subtle patterns present in the deception data. To incorporate cognitive load into deception detection, the extracted cognitive load features are concatenated with the final output of the AVPEF module for deception features. The final feature $F_{\mathrm{final}}$ is defined as the concatenation of $F_C$ and $F_D$.

The final feature $F_{\mathrm{final}}$ is then fed into a classifier network for the final deception detection prediction. The deception classifier consists of two linear layers.

### 3.3.2 OBJECTIVE FUNCTION

In deception detection, deception is inherently more challenging to detect than truth, as it involves greater cognitive load, leading to inconsistencies and complex behavioral patterns, which are often masked by strategic actions, making accurate identification difficult (DePaulo et al., 2003a; Ekman, 2009; Vrij, 2008; Vrij & Granhag, 2012; Zuckerman, 1981). This contrast underscores the need for a sophisticated approach capable of capturing the subtle differences between truth and deception.

In light of the inherent difficulty in distinguishing deception, we employ focal loss as the objective function to optimize model performance. Focal loss mitigates the impact of easily classified instances, allowing the model to prioritize more difficult cases, such as deception, which often involve subtle, complex patterns (Ross & Dollár, 2017). The focal loss is as follows:

$$L_{total} = -\sum_{i=1}^{N} y_i \log(p_i) - \sum_{i=1}^{N} (1 - p_i)^{\gamma} y_i \log(p_i) \tag{5}$$

Here, $p_i$ is the model's predicted probability for the true class. $\gamma$ is the focusing parameter that controls the contribution of hard-to-classify deceptive samples. This function enhances the model's ability to focus on deceptive instances that are often masked by strategic behaviors and varying cognitive loads. By incorporating focal loss, we ensure that our model captures subtle distinctions between truth and deception, improving classification performance on deceptive data.

## 4 EXPERIMENTS AND RESULTS

### 4.1 DATASET

**DOLOS (Guo et al., 2023)**. The DOLOS dataset is a large game show deception detection dataset, featuring rich deceptive conversations. This dataset is collected from a British reality comedy game show. It includes 1,675 video clips featuring 213 participants (141 male and 72 female). For each episode, video clips are extracted based on specific criteria: participants must speak only relevant content (i.e., telling the truth or lies) clearly without significant background noise, and their faces must be visible without occlusion. From 84 episodes, 1,675 clips, ranging from 2 to 19 seconds, were chosen. However, due to some clips becoming unavailable, the final dataset contains 1,656 clips. Each clip is annotated with a MUMIN coding scheme, focusing on non-verbal deceptive cues.

**Real Life Trail (Pérez-Rosas et al., 2015a)**. The Real Life Trail (RLT) dataset includes testimonies from defendants or witnesses in real court trials, consisting of 121 video clips with 61 deceptive and 60 truthful segments. The average duration of the clips is 28.0 seconds, with deceptive and truthful clips averaging 27.7 and 28.3 seconds, respectively. The speakers consist of 21 female and 35 male speakers, with ages ranging from 16 to 60 years. This comprehensive collection offers a robust resource for studying the nuances of deceptive and truthful behavior in high-stakes, real-world scenarios, providing a valuable benchmark for evaluating deception detection methodologies. Due to availability constraints, 110 out of the 121 clips were used in this study.

**Box of Lies (Soldner et al., 2019)**. The Box of Lies (BOL) dataset, derived from 'The Tonight Show Starring Jimmy Fallon,' features 25 video clips totaling 2 hours and 24 minutes. Each clip averages 6 minutes, with about three game rounds involving different guests and Jimmy Fallon. The dataset contains 1,049 utterances, of which 862 are deceptive and 187 truthful, with annotations focusing on verbal and non-verbal behaviors, including facial expressions and conversational cues, based on the MUMIN coding scheme. Details on statistical analysis can be found in Appendix A.4

### 4.2 DATA PREPROCESSING

For all datasets, including AVCAffe, DOLOS, RLT, and BOL, we applied the same data pre-processing pipeline. In each video, we uniformly selected 64 frames, applying the MTCNN (Zhang et al., 2016) face detector to isolate the facial regions. The extracted face images were resized to 160x160 pixels and subsequently normalized. For the audio, speech signals were resampled to ensure the W2V2 feature extractor produced 64 tokens. Additionally, we utilized the Demucs (Copet et al., 2024; Défossez et al., 2019) model to separate speech from background noise, minimizing the impact of background sounds and ensuring that only the speech component was used for further processing. Details on dataset preprocessing can be found in Appendix A.5

Table 1: Comparison of cognitive load prediction models based on parameters (Millions) and F1 scores across TLX subscales (M: Mental Demand, E: Effort, T: Temporal Demand). We compare the results with one of the representative approaches (Sarkar et al., 2023), which used audio (A) and visual (V) modalities. Sarkar et al. (2023) used combinations of VGG and ResNet for the audio modality, while the visual modality explored architectures such as ResNet3D, R(2+1)D, and MC3.

| Audio | Visual | # Parameters (Millions) | M | E | T |
|---|---|---|---|---|---|
| VGG16 | - | 138.5 | 58.8 | - | - |
| - | R(2+1)D-18 | 33.3 | 60.5 | - | - |
| VGG16 | ResNet3D-18 | 172.1 | 65.0 | - | - |
| VGG16 | - | 138.5 | - | 58.8 | - |
| - | R(2+1)D-18 | 33.3 | - | 65.5 | - |
| ResNet18 | R(2+1)D-18 | 43.5 | - | 60.8 | - |
| ResNet18 | - | 11.4 | - | - | 58.2 |
| - | MC3-18 | 11.6 | - | - | 60.0 |
| ResNet18 | ResNet3D-18 | 45.4 | - | - | 61.2 |
| **AVPEF** | | 71.2 (Trainable **5.2**) | 63.6 | 61.5 | 58.2 |

## 4.3 EXPERIMENT DETAIL

To investigate the impact and combinations of cognitive load features on deception detection, denoted as $F_{final}$, seven different feature sets are used: mental demand (M), effort (E), temporal demand (T), mental demand + effort (M + E), mental demand + temporal demand (M + T), effort + temporal demand(E + T), and mental demand + effort + temporal demand (M + E + T). These features are concatenated with the final output of the deception detection AVPEF network for experimental evaluation. The dimensions of each feature combination are standardized to 256, consistent with the dimensionality of the AVPEF network's feature.

For cognitive load feature extraction, we conduct experiments using the AVCAffe dataset, following the data-splitting strategy described in 3.2.1 to ensure no information leakage between training and validation sets ensuring that no information leakage occurred between the training and validation sets. The model was trained with a learning rate of 3e-4, a batch size of 8, and cross-entropy loss as the objective function. Four encoders were used, and training was carried out for 20 epochs with the Adam optimizer. An early stopping mechanism based on the F1-score was applied, halting training if no improvement was observed for 5 consecutive epochs.

For the deception detection framework, the average values of ACC, F1-score, and AUC are measured based on the 3-folds defined by the train-test protocol in the DOLOS dataset. The experiments are conducted over 20 epochs using the Adam optimizer, with an initial learning rate set to 1e-4. The batch size is 16, and focal loss (FL) with $\gamma = 2$ is used as the objective function. Additionally, the model architecture includes 4 encoders with a dropout rate of 0.5. The learning rate is adjusted using the StepLR scheduler, where the learning rate is halved every 5 epochs. All experiments are performed in a Python 3.8.19 and PyTorch 1.13.1 environment.

## 4.4 RESULT OF COGNITIVE LOAD PREDICTION

Table 1 presents the top-performing backbone combinations for each TLX subscale (Mental Demand, Effort, and Temporal Demand), evaluated using the F1-score. It includes the total and trainable parameters for each model, with the parameters of AVPEF included for reference. Despite having significantly fewer trainable parameters, AVPEF achieves competitive F1-scores, highlighting its effectiveness and parameter efficiency in predicting cognitive load across multiple dimensions.

## 4.5 RESULTS ON DOLOS DATASET

The baseline model is trained using AVPEF on the DOLOS dataset, incorporating MUMIN features for multi-task learning. The model uses cross-entropy loss and a four-layer encoder. The performance of the AVPEF module combined with cognitive load features on the DOLOS dataset is presented in Table 2. For all seven combinations of cognitive load features, our AVDDCL outperforms the baseline on the DOLOS dataset. Notably, the proposed method achieves superior performance

compared to the multi-task approach using MUMIN (Allwood et al., 2005), demonstrating the feasibility of automated deception detection without human labeling through the integration of cognitive load features. An analysis of the individual cognitive load features reveals that mental demand achieves the highest performance, followed by temporal demand and effort.

Notably, the best overall performance is obtained when all three cognitive load features are combined, highlighting the value of feature integration for deception detection. The cognitive load features are not independent of each other but are related. By leveraging the combination of these features, the proposed method effectively captures subtle interactions across cognitive load aspects, providing a robust framework for automated deception detection. This finding underscores the significance of adopting a comprehensive approach to modeling intricate human behaviors, as focusing on isolated features can risk overlooking the multifaceted nature of the cognitive processes involved in deceptive actions.

Table 2: The deception detection performance on the DO-LOS dataset; AVDDCL: Audio-Visual Deception Detection with Cognitive Load. Various combinations of these cognitive load features are evaluated and compared with existing benchmark results. The metrics are ACC (%), F1-score (%), and AUC (%).

| Method (w/ features) | ACC | F1 | AUC |
|---|---|---|---|
| Guo et al. (2023) | 64.8 | 71.2 | 62.7 |
| Guo et al. (2023) (Multi) | 66.8 | 73.4 | 64.6 |
| AVDDCL (M) | 67.7 | 73.0 | 66.3 |
| AVDDCL (E) | 63.0 | 70.5 | 60.7 |
| AVDDCL (T) | 67.0 | 72.8 | 65.3 |
| AVDDCL (M+E) | 66.7 | 71.9 | 65.3 |
| AVDDCL (M+T) | 67.4 | 73.3 | 65.6 |
| AVDDCL (E+T) | 66.1 | 71.4 | 64.6 |
| AVDDCL (M+E+T) | **68.0** | **73.4** | **66.5** |

## 4.6 EXPERIMENTS ON DIVERSE DATASETS FOR GENERALIZATION

We evaluate the generalization capabilities of the AVDDCL through a series of experiments on high-stakes (RLT) and low-stakes (BOL) deception datasets. High-stakes deception, which carries severe consequences such as legal penalties, differs significantly from low-stakes deception, where the effects are minimal (Porter & ten Brinke, 2010; Wright Whelan et al., 2015). This disparity introduces challenges for models attempting to generalize between the two contexts.

Table 3: Within-dataset experiments on RLT (high-stakes) and BOL (low-stakes) datasets. Metrics are ACC (%), F1-score (%), and AUC (%) over 5-fold cross-validation.

| Model | Modality | Train RLT / Test RLT | | | Train BOL / Test BOL | | |
|---|---|---|---|---|---|---|---|
| | | ACC | F1-score | AUC | ACC | F1-score | AUC |
| Camara et al. (2024) | V | $62.7 \pm 1.0$ | $62.8 \pm 1.0$ | $64.1 \pm 1.1$ | $62.4 \pm 1.2$ | $58.2 \pm 2.2$ | $62.4 \pm 1.2$ |
| Guo et al. (2023) | V + A | $82.7 \pm 1.3$ | $83.3 \pm 1.2$ | $82.8 \pm 1.3$ | $67.0 \pm 1.6$ | $62.1 \pm 2.4$ | $67.0 \pm 1.4$ |
| **AVDDCL (Ours)** | V + A | $\mathbf{86.4 \pm 1.4}$ | $\mathbf{85.6 \pm 1.6}$ | $\mathbf{86.6 \pm 1.4}$ | $\mathbf{74.0 \pm 1.9}$ | $\mathbf{68.5 \pm 3.5}$ | $\mathbf{74.0 \pm 1.9}$ |

Table 4: Cross-corpus experiments on RLT (high-stakes) and BOL (low-stakes) datasets. The metrics are ACC (%), F1-score (%), and AUC (%).

| Model | Modality | Train RLT / Test BOL | | | Train BOL / Test RLT | | |
|---|---|---|---|---|---|---|---|
| | | ACC | F1-score | AUC | ACC | F1-score | AUC |
| Camara et al. (2024) | V | 42.8 | 39.8 | 50.9 | 50.2 | 46.6 | 49.7 |
| Biçer & Dibeklioğlu (2023) | V | - | 44.1 | - | - | 44.7 | - |
| Biçer & Dibeklioğlu (2023) | A | - | 38.2 | - | - | 45.6 | - |
| Guo et al. (2023) | V + A | 48.6 | 15.7 | 48.6 | 52.6 | 58.8 | 53.2 |
| **AVDDCL (Ours)** | V + A | 47.4 | 7.1 | 47.3 | **55.3** | 57.6 | **55.5** |

To evaluate AVDDCL's performance, we employed 5-fold cross-validation to ensure robustness and reduce the influence of data splits on the results. We conducted both within-dataset and cross-corpus experiments, using accuracy, F1-score, and AUC as evaluation metrics. For the BOL dataset, data preparation involved organizing utterances into rounds and applying under-sampling to the 'deception' class to address the class imbalance issue, resulting in a dataset of 128 videos. Comprehensive data preparation steps and related statistics are detailed in Appendix A.4. AVDDCL showed significant performance within datasets, achieving 86.4% accuracy for high-stakes and 74.0% for low-stakes deception scenarios (Table 3). However, cross-dataset evaluations revealed variations in performance due to domain differences between high-stakes and low-stakes deception. When trained

on BOL and evaluated on RLT, the model reached an accuracy of 55.3%, suggesting some degree of pattern transferability between datasets. In contrast, training on RLT and testing on BOL resulted in a lower accuracy (47.4%), likely due to the more subtle cues associated with low-stakes deception, as high-stakes scenarios generally involve more pronounced deceptive indicators compared to the nuanced patterns in low-stakes contexts (Wright Whelan et al., 2014). These results highlight the challenges posed by domain differences in deception detection. While AVDDCL demonstrates robust performance in within-dataset evaluations, it also shows promise in improving generalization from low-stakes to high-stakes scenarios through the integration of cognitive load features. Future work is required to focus on refining the model to better capture subtle cues in low-stakes deception, aiming to enhance cross-domain performance and generalize across diverse deception contexts.

### 4.7 ABLATION STUDY

#### 4.7.1 IMPACT OF FOCAL LOSS AND GAMMA TUNING

To validate the effectiveness of focal loss, we conducted a comprehensive evaluation across a wide range of $\gamma$ using the DOLOS dataset. We included $\gamma=0$ as a baseline, equivalent to Cross Entropy Loss, to assess the impact of focal loss on model performance. Table 5 shows that focal loss consistently outperformed cross-entropy loss, leading to improved metrics across the board. Moreover, the model achieved the highest Accuracy, F1-score, and AUC when $\gamma=2$, which was selected as the optimal parameter for our analyses.

Table 5: AVDDCL(M+E+T) ablation study and hyperparameter tuning on DOLOS dataset. The metrics are ACC(%), F1(%), and AUC(%). Note that C/E represents Cross Entropy Loss.

| Method | $\gamma$ | ACC | F1 | AUC |
|---|---|---|---|---|
| AVDDCL (M+E+T) | 0 (C/E) | 66.2 | 72.5 | 64.3 |
| AVDDCL (M+E+T) | 0.5 | 66.8 | 72.3 | 65.4 |
| AVDDCL (M+E+T) | 1 | 65.3 | 70.4 | 64.1 |
| AVDDCL (M+E+T) | 1.5 | 66.6 | 72.6 | 64.5 |
| AVDDCL (M+E+T) | 2 | **68.0** | **73.4** | **66.5** |
| AVDDCL (M+E+T) | 3 | 67.5 | 73.3 | 65.7 |
| AVDDCL (M+E+T) | 5 | 66.3 | 70.7 | 65.5 |

This improvement highlights the effectiveness of focal loss in addressing the complexities of deception detection, enabling the model to focus on subtle cognitive and behavioral patterns that differentiate deception from truth. This aligns with existing research emphasizing the cognitive demands and behavioral inconsistencies inherent in deceptive actions (Vrij, 2008; Ekman, 2009).

#### 4.7.2 VISUALIZATION OF EXTRACTED FEATURE

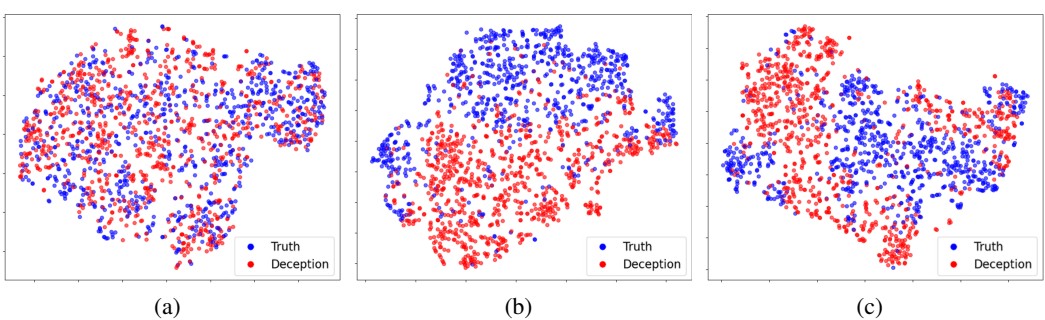

Figure 3: Visualization of audio-visual features using t-SNE: (a) Cognitive load feature, (b) Deception feature, (c) Cognitive + Deception feature. Reds indicate deception, blues indicate truth.

To gain further insights into the behavior of our AVDDCL model, we conduct t-SNE (Van der Maaten & Hinton, 2008) visualizations for different feature sets extracted from the audio-visual data. Figure 3 shows the visualization of audio-visual features from our AVDDCL model. Figure 3(a), Figure 3(b), and Figure 3(c) depict the cognitive load features, deception features, and the integrated features of cognitive load features and deception features, respectively.

In Figure 3(c), the combination of cognitive load and deception-specific features provides a more refined differentiation between truth and deception. This result showcases not just a clear-cut bound-

ary but a broader variety of how deception manifests. The additional cognitive load context, derived from mental demand, temporal load, and effort, enriches the model's ability to handle more complex patterns of deceptive behavior. This expanded separation, rather than simplifying the distinction, highlights the varied and layered nature of deception, allowing the model to capture multiple aspects of both cognitive and behavioral cues in deceptive instances.

This variation in observed patterns suggests the presence of distinct dimensions within deceptive behaviors, supporting findings that deception is not a singular phenomenon but a complex and multi-layered construct (Hartwig & Bond Jr, 2011; Porter & ten Brinke, 2010; Sporer & Schwandt, 2007). Future research could integrate cognitive load and deception features to explore the diverse forms of deception, enabling systems to categorize different deceptive strategies. This refined approach could help in developing models that not only distinguish between truth and deception but also classify deception into various subtypes, each characterized by distinct cognitive efforts and behavioral traits, offering a deeper and more detailed analysis of deceptive behavior.

## 5  DISCUSSION AND FUTURE WORK

Existing studies emphasize that fully automated decision-making systems, while significantly efficient, can raise concerns considering fairness and reliability when human judgment is excluded (Kern et al., 2022). It is important in high-stakes scenarios, such as law enforcement or medical diagnosis, where automation bias can lead to over-reliance on or disregard for system outputs (Belavadi et al., 2020). In these contexts, false positives can carry substantial risks, underlining the need for balanced and transparent systems. Misuse of automated systems may also erode public trust, enable manipulation, spread misinformation, or perpetuate discrimination against marginalized groups (Biçer & Dibeklioğlu, 2023). Therefore, rigorous ethical considerations are crucial before implementing such systems. Incorporating human-in-the-loop systems is essential to contributing to fairness and efficiency, leveraging human judgment alongside automated capabilities (Cummings, 2017; Khan et al., 2021)

Integrating human cognitive and psychological factors into system design can greatly enhance the trustworthiness of models (Cummings, 2017; Mosier & Skitka, 2018). Building on this principle, we integrate cognitive load-based features to address automation bias and ensure more reliable decision-making. However, unresolved biases in datasets and demographic factors induced during training pose challenges to generalization across diverse contexts. The lack of real-world datasets, such as the Real-Life Trial dataset, exacerbates domain-specific overfitting, limiting the model's applicability to both high-stakes and low-stakes deception scenarios (King & Neal, 2024). In addition, demographic factors like gender, age, and cultural differences significantly influence deception cues, impacting model performance (Abouelenien et al., 2018; Levitan et al., 2016; Naven et al., 2020).

Future research should focus on creating more diverse datasets and enhancing domain generalization to improve cross-context performance. Incorporating human-in-the-loop systems will be key to ensuring fairness and reliability in automated deception detection. Improving explainability through interpretable decision-making processes is also essential for building trust in high-stakes contexts. Bias mitigation strategies, such as fairness-aware training and demographic balancing, are crucial for ensuring equitable outcomes. Moreover, making the parameters $\alpha$ and $\gamma$ in focal loss adaptive and learnable could further enhance the model's ability to generalize across diverse, real-time contexts. Furthermore, optimizing models for real-time applications through pruning and quantization can minimize latency and computational demands while balancing efficiency and reliability for robust performance in real-world settings.

## 6  CONCLUSION

We propose the AVDDCL framework, a novel approach to automated deception detection using cognitive load features extracted from audio-visual data. Our findings show that incorporating multiple cognitive load dimensions significantly improves model performance, supporting the idea that cognitive load and its dimensions are key to detecting deception. The use of focal loss further enhances the model by targeting difficult instances, boosting accuracy and robustness. This scalable, automated solution eliminates the need for specialized equipment or human annotations, enabling more reliable and practical real-world applications.

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

## A    APPENDIX

### A.1    DETAILS ABOUT UT-ENCODER

The UT-Encoder (UTE) refers to the entire architecture that incorporates the W2V2 or ViT encoder alongside UT-Adapters. UT-Adapters are placed in parallel with MHSA and feed-forward layer (FFD) in each encoder of W2V2 and ViT. The encoder components, including MSHA, Multi-Layer Perceptron(MLP), and normalization layers, remain frozen, while only the UT-Adapter parameters are learnable. Each UT-Adapter consists of a series of linear layers and 1D-convolutional layers. The architecture of the UT-Adapter is as follows:

$$U(X) = L_2 \left( P \left( C \left( P \left( L_1(X; W_1) \right); W_C \right) \right); W_2 \right) \tag{6}$$

Here, $L_1$ and $L_2$ represent the Linear 1 and Linear 2 layers, respectively, and $P$ and $C$ denote the permutation and 1D-convolutional Layers. The weights $W_1 \in \mathbb{R}^{D \times 128}$ and $W_2 \in \mathbb{R}^{128 \times D}$ are trainable parameters of the linear layers $L_1$ and $L_2$, while $W_C$ is the trainable weight for the 1D-convolutional layer with a kernel size of 3. Specifically, $L_1$ projects the input $X \in \mathbb{R}^{L \times D}$ to $X \in \mathbb{R}^{L \times 128}$. The Permutation layer $P$ shifts the data from $X \in \mathbb{R}^{L \times 128}$ to $X \in \mathbb{R}^{128 \times L}$. The convolutional layer $C$ is then applied along the temporal dimension to capture temporal dynamics. After the convolution operation, the Permutation layer and $L_2$ project the data back from $X \in \mathbb{R}^{128 \times L}$ to $X \in \mathbb{R}^{L \times D}$.

UTE block effectively utilizes the UT-Adapter to capture local temporal information, while the MHSA and MLP modules focus on learning global temporal and spatial attention. This architecture allows the model to balance parameter efficiency and performance.

### A.2    DEATILS ABOUT AUDIO-VISUAL FUSION

To facilitate fusion based on the interaction between Audio and Visual data, the output features from both modalities are initially projected into a lower-dimensional embedding space to reduce computational costs. After projecting the inputs, the PAVF module calculates the cross-modal correlation matrix $P_i$ using a trainable weight matrix $W_p$ as follows:

$$P_i = X_a' W_P (X_v')^\top \tag{7}$$

where $X_a'$ and $X_v'$ are the reduced-dimensional representations of the audio and visual encoder outputs, respectively. The cross-modal correlation matrix $P_i$ indicates the importance of the interactions between specific audio and visual sequences, which is crucial for tasks like deception detection.

The module then applies cross-modal attention to both modalities, refining the feature representations as:

$$\tilde{X}_v = \text{Softmax}(P_i) X_v + X_v, \quad \tilde{X}_a = \text{Softmax}(P_i^\top) X_a + X_a, \tag{8}$$

The attended features, $\tilde{X}_v$ and $\tilde{X}_a$ are concatenated to form a joint representation:

$$\tilde{X}_{va} = \tilde{X}_v \oplus \tilde{X}_a \tag{9}$$

which is then processed through a fusion head comprising linear projection, normalization, and ReLU activation:

$$\tilde{X}_{va} = \text{ReLU}(\text{LN}(L_p(\tilde{X}_{va}))) \tag{10}$$

### A.3 Details about Cross Entropy Loss and Focal Loss

As mentioned in subsection 3.3.2, due to the inherent challenges in detecting deception, which often requires handling more subtle and complex patterns, we utilize the focal loss, a function that builds upon cross-entropy loss. Cross entropy loss is defined as follows:

$$CE(p, y) = \begin{cases} -\log(p) & \text{if } y = 1 \\ -\log(1 - p) & \text{otherwise.} \end{cases} \tag{11}$$

define $p_t$:

$$p_t = \begin{cases} p & \text{if } y = 1 \\ 1 - p & \text{otherwise.} \end{cases} \tag{12}$$

and can rewrite

$$CE(p, y) = CE(p_t) = -\log(p_t). \tag{13}$$

In the given equation, $y \in \{\pm 1\}$ indicates the actual class label, while $p \in [0, 1]$ represents the model's estimated probability for the class where $y = 1$.

However, one property of CE loss is even well-classified examples continue to contribute a substantial portion to the overall loss. When these relatively minor losses are aggregated across a large number of easy examples, they can disproportionately diminish the influence of rarer, more challenging classes.

A simple method to address class imbalance is to introduce a weighting factor, $\alpha \in [0, 1]$, as a hyperparameter for class 1 in CE, and $1 - \alpha$ for class -1 which can be expressed as follows:

$$CE(p_t) = -\alpha_t \log(p_t) \tag{14}$$

While $\alpha$ balances the importance of positive/negative examples, it does not differentiate between easy/hard examples. Therefore, by adding the modulating factor $-(1 - p_t)^\gamma$ with tunable focusing parameter $\gamma \geq$, the objective function is restructured to down-weight easy examples and focus on the hard negatives as follows:

$$FocalLoss(p_t) = -\alpha_t (1 - p_t)^\gamma \log(p_t) \tag{15}$$

## A.4 DATA STATISTICS

To ensure a fair and rigorous evaluation of AVDDCL's performance, we employed a 5-fold cross-validation strategy for both the RLT and BOL datasets. This approach ensured robust results by minimizing the effects of data splits and allowed for consistent comparison across different experimental settings.

The original BOL dataset required specific preprocessing steps to align with the structure of the RLT dataset and address its inherent class imbalance. First, the BOL dataset, which initially featured utterance-based labeling, was adjusted by grouping utterances into "rounds" to reduce discrepancies with the video-based labeling format of the RLT dataset. This grouping reflected the gameplay structure observed in each video. Second, the significant imbalance between the 'deception' and 'truth' classes in the BOL dataset, with an overrepresentation of deceptive samples, was addressed by applying random under-sampling of the 'deception' class. This balancing ensured that the model could effectively learn both classes without bias.

After these preprocessing steps, the balanced BOL dataset consisted of 128 videos, while the RLT dataset included 110 videos. Detailed statistics of the preprocessed datasets, including the distribution of deceptive and truthful instances, are summarized in Table 6.

Table 6: Comparison of DOLOS, Real Life Trial(RLT), and Box of Lies(BOL) Dataset Statistics.

| Dataset | Total Files | Avg Duration (s) | Std Dev (s) | # Deception | # Truthful |
|---|---|---|---|---|---|
| DOLOS | 1,656 | 5.4 | 4.7 | 886 | 690 |
| Real Life Trial | 110 | 28.0 | 13.3 | 53 | 57 |
| Box of Lies | 128 | 19.4 | 21.2 | 64 | 64 |

## A.5 DATA PRE-PROCESSING

### A.5.1 VIDEO PRE-PROCESSING

For all datasets, the video preprocessing pipeline involved a series of steps to extract meaningful facial regions from video frames while maintaining temporal consistency.

Frames were sampled uniformly from each video at a fixed rate of 20 frames per second (FPS). This was implemented using OpenCV, where the native FPS of each video was determined using the `cv2.VideoCapture` function and its `CAP_PROP_FPS` property. For videos with an FPS greater than 20, the interval between sampled frames was calculated as `frame_interval = max(1, int(fps / 20))`, ensuring an even distribution of frames. If the FPS was lower than 20, no frames were skipped to preserve the temporal resolution.

Once frames were extracted, facial regions were detected using MTCNN. The MTCNN model provided bounding box coordinates and facial keypoints such as the positions of the eyes, nose, and mouth. For frames containing multiple faces, a tracking mechanism was employed to assign unique IDs to each detected face. This mechanism compared bounding box positions and used a Euclidean distance threshold of 40 pixels between nose keypoints to determine if a detected face matched an existing ID. Across all frames in a video, the face with the highest frequency of detection was selected as the primary face for further processing.

The selected facial regions were cropped using the bounding box coordinates provided by MTCNN and resized to a fixed resolution of 160×160 pixels using OpenCV's `cv2.resize` function. Bilinear interpolation was applied during resizing to preserve facial details.

Finally, 64 frames were uniformly sampled from each video to ensure consistent representation across datasets.

### A.5.2 AUDIO PRE-PROCESSING

The audio preprocessing pipeline was designed to extract and refine speech signals from the videos while minimizing background noise.

The first step involved extracting audio tracks from each video using the MoviePy library. The `VideoFileClip` class was used to load each video, and its audio component was accessed and exported as a `.wav` file. This ensured that the audio data retained the original fidelity of the recording. Videos without audio tracks were logged and excluded from further processing to maintain pipeline integrity.

To further refine the audio signals, we applied the Demucs model (Défossez et al., 2019), a deep learning-based source separation model. The Demucs model was configured to separate the audio into two components: vocals (speech) and non-vocals (background noise). Each `.wav` file was processed using the two-stem configuration, which focuses on isolating speech content from other sound elements.

By combining noise separation and structured organization, the preprocessing pipeline provided clean and consistent speech signals for subsequent feature extraction and analysis. This ensured that the audio data was of high quality and aligned with the temporal structure of the video data.

A.6    EVALUATING COGNITIVE LOAD SUBSCALES IN RELATION TO DECEPTION LABELS

Table 7: Confusion Matrix for Cognitive Load Subscale Classification and Deception Labels

| Cognitive Load Subscale | Deception/Truth | Low | High |
|---|---|---|---|
| **Mental Demand** | Deception | 77 | 811 |
| | Truth | 71 | 697 |
| **Effort** | Deception | 547 | 341 |
| | Truth | 448 | 320 |
| **Temporal Demand** | Deception | 156 | 732 |
| | Truth | 147 | 621 |

This section quantifies the relationship between NASA-TLX cognitive load subscales (mental demand, effort, and temporal demand) and deception labels. Using the pre-trained AVPEF(see 3.2.1) with linear layers intact, we classified each subscale as high or low and compared these classifications to deception labels in the DOLOS dataset to evaluate their contributions to deception detection accuracy.

The confusion matrix results (Table 7) reveal that individual TLX subscales have weak correlations with deception labels, as indicated by Cramér's V values below 0.1. However, as demonstrated in our findings in 4.5 combining multiple subscales significantly enhances deception detection performance. This aligns with prior research suggesting that TLX subscales interact rather than operate independently  (Nikulin et al., 2019).

These findings highlight the importance of interactions between cognitive load subscales in capturing the complexities of deception. Future work will explore these interdependencies further using larger datasets and advanced models to enhance the framework's effectiveness.

