# OpenReview forum: "Enhancing Deception Detection with Cognitive Load Features: An Audio-Visual Approach"
_ICLR.cc/2025/Conference — ICLR 2025 Conference Withdrawn Submission_

### Official Review · Reviewer_rGQ1 · 2024-10-16

**Soundness:** 3
**Presentation:** 3
**Contribution:** 3
**Rating:** 6
**Confidence:** 4

**Summary:**

This paper propose a framework to automatically extracting cognitive load from audio-visual data, and integrates them into a deception deception pipeline to enchance its robustness.

**Strengths:**

- Novel idea of pretraining of model on AVCAffe dataset to extract cognitive load features in order to support downstream deception detection task.
- Novel Audio-Visual Parameter-Efficient Fusion network (AVPEF) and training loss used. An innovative network architecture was proposed to fuse both audio and visual modality with parameter efficient finetuing. The use of focal loss to put emphasis on difficult samples.

**Weaknesses:**

- There is insufficient evidence to show that AVPEF is useful. Results from Table 1 suggest that other architectures outperforms the proposed AVPEF. Also, the lack of comparison against other multimodal learning fusion methods makes it difficult to determine the effectiveness of AVPEF.
- There is a lack of considerations on ethics and bias. More exploration into the biases of their model issues will make this study more comphresive and as well as to ensure responsible deployment of such models.

**Questions:**

- **line 295** Wrong citation of DOLOS dataset.
- Addressing the concerns stated in the weakness.

**Details Of Ethics Concerns:**

Potential biases needs to be address for deception detection since it will be use in high stakes settings such as security, legal trials and job interviews.

---

> ### Author Response · Authors · 2024-11-20
> **Response to Reviewer rGQ1 [1/5]**
>
> Thank you for your valuable feedback and insightful suggestions. We truly appreciate your thorough review and hope that the following responses address your concerns and clarify any uncertainties.
>
> We also welcome any further comments from the reviewer and will make every effort to incorporate them.
>
> **Weakness 1: Concerns about the lack of evidence supporting the proposed AVPEF's usefulness (Table 1).**
>
> We sincerely thank the reviewer for highlighting the need to provide additional evidence supporting the effectiveness of AVPEF. To clarify, AVPEF is not a novel contribution of this study but an existing feature extraction approach chosen for its parameter efficiency and capability to model temporal information—both of which are critical for cognitive load prediction.
>
> The motivation and effectiveness of using AVPEF are outlined as follows:
>
> - **Parameter Efficiency**: AVPEF demonstrates remarkable parameter efficiency, utilizing only 5.2M trainable parameters—significantly fewer than many other multimodal or unimodal feature extractors, which often require tens of millions of parameters. Unlike other models that exhibit identical values for total and trainable parameters, AVPEF’s reduced parameter count minimizes computational costs and enhances efficiency, particularly in resource-constrained environments. By leveraging pre-trained ViT and W2V2 models with a uniform temporal adapter, AVPEF achieves effective learning across diverse settings while maintaining low computational demands.
>
> - **Importance of Temporal Information**: Cognitive load is inherently linked to temporal dynamics, as supported by several studies [1,2,3]. AVPEF, equipped with a uniform temporal encoder, excels in capturing temporal variations in cognitive load states. This capability enables AVPEF to measure cognitive load with high accuracy, overcoming limitations in baseline models that may neglect critical temporal dependencies.
>
> - **Baseline Inconsistencies**: Our analysis reveals that baseline models often exhibit inconsistent performance across different modalities and TLX subscales. The optimal choice of feature extractors frequently varies for each TLX subscale, leading to experimental complexity and reduced generalizability. In contrast, AVPEF provides stable and reliable performance across both modalities and subscales, ensuring consistency in its predictions.
>
> - **Comparable Performance**: Despite its reduced parameter count, AVPEF achieves F1 scores that are comparable to other models for TLX subscales, including Mental Demand, Effort, and Temporal Demand. This combination of efficiency and accuracy positions AVPEF as a practical and robust option for cognitive load prediction.
>
> The table below shows a comparison of cognitive load prediction models based on parameters (Millions) and F1 scores across TLX subscales (M: Mental, E: Effort, T: Temporal).
> | Audio     | Visual         | \# Parameters (Millions) | M    | E    | T    |
> |-----------|----------------|--------------------------|------|------|------|
> | VGG16     | -              | 138.5                    | 58.8 | -    | -    |
> | -         | R(2+1)D-18     | 33.3                     | 60.5 | -    | -    |
> | VGG16     | ResNet3D-18    | 172.1                    | 65.0 | -    | -    |
> | VGG16     | -              | 138.5                    | -    | 58.8 | -    |
> | -         | R(2+1)D-18     | 33.3                     | -    | 65.5 | -    |
> | ResNet18  | R(2+1)D-18     | 43.5                     | -    | 60.8 | -    |
> | ResNet18  | -              | 11.4                     | -    | -    | 58.2 |
> | -         | MC3-18         | 11.6                     | -    | -    | 60.0 |
> | ResNet18  | ResNet3D-18    | 45.4                     | -    | -    | 61.2 |
> | **AVPEF** |                | **71.2 (Trainable 5.2)**  | 63.6 | 61.5 | 58.2 |

---

> ### Author Response · Authors · 2024-11-20
> **Response to Reviewer rGQ1 [2/5]**
>
> **Weakness 1: Concerns about the lack of evidence supporting the proposed AVPEF's usefulness (Table 1).**
>
> We sincerely thank the reviewer for emphasizing the importance of comparing AVDDCL with other multimodal fusion methods. While the primary goal of AVDDCL is not to introduce new fusion techniques, we have nevertheless compared its performance with existing baselines, as shown in the table below. These results demonstrate that AVDDCL outperforms the baseline models on both the RLT and BOL datasets, highlighting its robustness and adaptability across diverse deception detection contexts.
>
> Note that, for consistency in evaluation, all multimodal setups (Visual + Audio) utilized the same AVPEF architecture for feature extraction. The focus of this study is to validate the effectiveness of AVDDCL in leveraging cognitive load features for deception detection, rather than to benchmark against a variety of multimodal fusion strategies. However, we acknowledge the value of such comparisons and plan to incorporate additional fusion techniques in future work to further substantiate the contributions of AVDDCL.
>
> | **Model**            | **Modality** | **Train RLT / Test RLT**       |                        |             | **Train BOL / Test BOL**       |                        |             |
> |----------------------|--------------|--------------------------------|------------------------|-------------|--------------------------------|------------------------|-------------|
> |                      |              | **ACC**                       | **F1-score**           | **AUC**     | **ACC**                       | **F1-score**           | **AUC**     |
> | Camara et al. (2024) [4] | V            | 62.7 ± 1.0                         | 62.8 ± 1.0                | 64.1 ± 1.1        | 62.4  ± 1.2                         | 58.2  ± 2.2                 | 62.4  ± 1.2       |
> | Guo et al. (2023) [5]  | V + A        | 82.7 ± 1.3                         | 83.3 ± 1.2                 | 82.8 ± 1.3        | 67.0  ± 1.6                         | 62.1  ± 2.4                 | 67.0  ± 1.2       |
> | **AVDDCL (Ours)**    | V + A        | **86.4 ± 1.4**                      | **85.6 ± 1.6**              | **86.6 ± 1.4**    | **74.0 ± 1.9**                      | **68.5 ± 3.5**              | **74.0 ± 1.9**    |
>
> The revised and updated details regarding this topic can be found in **Section 4.3** and **Section 4.5**.
>
> [1] Puma, S., Matton, N., Paubel, P. V., & Tricot, A. (2018). Cognitive load theory and time considerations: Using the time-based resource sharing model. Educational Psychology Review, 30, 1199-1214.
>
> [2] Liu, Y., Yu, Y., Ye, Z., Li, M., Zhang, Y., Zhou, Z., ... & Zeng, L. L. (2023). Fusion of spatial, temporal, and spectral EEG signatures improves multilevel cognitive load prediction. IEEE Transactions on Human-Machine Systems, 53(2), 357-366.
>
> [3] Li, Y., Li, K., Wang, S., Wu, H., & Li, P. (2025). A spatiotemporal separable graph convolutional network for oddball paradigm classification under different cognitive-load scenarios. Expert Systems with Applications, 262, 125303.
>
> [4] Camara, M. K., Postal, A., Maul, T. H., & Paetzold, G. H. (2024). Can lies be faked? Comparing low-stakes and high-stakes deception video datasets from a Machine Learning perspective. Expert Systems with Applications, 249, 123684.
>
> [5] Guo, X., Selvaraj, N. M., Yu, Z., Kong, A. W. K., Shen, B., & Kot, A. (2023). Audio-visual deception detection: Dolos dataset and parameter-efficient crossmodal learning. In Proceedings of the IEEE/CVF International Conference on Computer Vision (pp. 22135-22145).

---

> ### Author Response · Authors · 2024-11-20
> **Response to Reviewer rGQ1 [3/5]**
>
> **Weakness 2: Ethical considerations and biases are not explored, especially for high-stakes applications.**
>
> We sincerely thank the reviewer for highlighting the critical importance of ethical considerations and biases, particularly in high-stakes applications.
>
> First, we would like to emphasize that the datasets used in this study—DOLOS, BOL, and RLT—are publicly available and widely used in research. Importantly, no sensitive or private data were included in the models.
>
> To address these concerns, we expanded the Discussion and Future Work section in the revised manuscript. Below, we summarize our discussion and the steps we propose to mitigate these challenges:
>
> - **Ethical Implications in High-Stakes Scenarios**: Fully automated systems, while efficient, can introduce automation bias and lead to over-reliance on system outputs when human judgment is excluded, especially in sensitive contexts such as law enforcement, medical diagnosis, or political decision-making[1,2]. In high-stakes situations, false positives can cause significant harm, underscoring the need for balanced and transparent systems. Moreover, the potential misuse of these systems—such as undermining public trust, spreading misinformation, or exacerbating discrimination against marginalized groups—presents substantial ethical challenges[3]. To address these concerns, we emphasize the importance of integrating human-in-the-loop systems, where human judgment complements automated efficiency, ensuring fairness, accountability, and transparency[4,5]. In addition, enhancing model explainability through interpretable decision-making processes is essential to building trust and accountability, particularly in high-stakes contexts.
>
> - **Addressing Dataset Bias and Demographic Diversity**: Incorporating human cognitive and psychological factors into system design can significantly improve model trustworthiness[4,6]. Building on this principle, our study integrates cognitive load-based features to address automation bias and promote reliable decision-making. However, unresolved dataset biases and demographic biases, such as those related to gender, age, and cultural differences, remain significant challenges in automated deception detection[7,8,9]. These biases hinder the model’s ability to generalize across diverse populations and contexts, limiting its real-world applicability. To address these challenges, future work will focus on expanding dataset diversity and implementing bias mitigation strategies, including fairness-aware training and demographic balancing. These efforts aim to ensure equitable performance across diverse groups and enhance the system’s robustness.
>
> By integrating these ethical considerations into the design and deployment of automated deception detection systems, we aim to develop solutions that are not only effective but also socially responsible and fair. These points reflect our commitment to addressing challenges related to fairness, transparency, and accountability, particularly in high-stakes applications.
>
> The revised and updated details regarding this topic can be found in **Section 5**.

---

> > ### Author Response · Authors · 2024-11-20
> > **Response to Reviewer rGQ1 [4/5]**
> >
> > **Weakness 2: Ethical considerations and biases are not explored, especially for high-stakes applications.**
> >
> > [1] Kern, C., Gerdon, F., Bach, R. L., Keusch, F., & Kreuter, F. (2022). Humans versus machines: Who is perceived to decide fairer? Experimental evidence on attitudes toward automated decision-making. Patterns, 3(10).
> >
> > [2] Belavadi, V., Zhou, Y., Bakdash, J. Z., Kantarcioglu, M., Krawczyk, D. C., Nguyen, L., ... & Thuriasingham, B. (2020, October). MultiModal deception detection: Accuracy, applicability and generalizability. In 2020 Second IEEE International Conference on Trust, Privacy and Security in Intelligent Systems and Applications (TPS-ISA) (pp. 99-106). IEEE.
> >
> > [3] Biçer, B., & Dibeklioğlu, H. (2023). Automatic Deceit Detection Through Multimodal Analysis of High-Stake Court-Trials. IEEE Transactions on Affective Computing.
> >
> > [4] Cummings, M. L. (2017). Automation bias in intelligent time critical decision support systems. In Decision making in aviation (pp. 289-294). Routledge.
> >
> > [5] Khan, W., Crockett, K., O'Shea, J., Hussain, A., & Khan, B. M. (2021). Deception in the eyes of deceiver: A computer vision and machine learning based automated deception detection. Expert Systems with Applications, 169, 114341.
> >
> > [6] Mosier, K. L., & Skitka, L. J. (2018). Human decision makers and automated decision aids: Made for each other?. In Automation and human performance (pp. 201-220). CRC Press.
> >
> > [7] Levitan, S. I., Levitan, Y., An, G., Levine, M., Levitan, R., Rosenberg, A., & Hirschberg, J. (2016, June). Identifying individual differences in gender, ethnicity, and personality from dialogue for deception detection. In Proceedings of the second workshop on computational approaches to deception detection (pp. 40-44).
> >
> > [8] Naven, G., Sen, T., Gerstner, L., Haut, K., Wen, M., & Hoque, E. (2020, November). Leveraging shared and divergent facial expression behavior between genders in deception detection. In 2020 15th IEEE international conference on automatic face and gesture recognition (FG 2020) (pp. 428-435). IEEE.
> >
> > [9] Abouelenien, M., Burzo, M., Pérez-Rosas, V., Mihalcea, R., Sun, H., & Zhao, B. (2018). Gender differences in multimodal contact-free deception detection. IEEE MultiMedia, 26(3), 19-30.

---

> ### Author Response · Authors · 2024-11-20
> **Response to Reviewer rGQ1 [5/5]**
>
> **Question 1: Address Line 295's incorrect citation of the DOLOS dataset.**
>
> We sincerely thank the reviewer for pointing out the incorrect citation of the DOLOS dataset. We have thoroughly reviewed and corrected the citation in the revised manuscript to ensure its accuracy. This correction ensures that the dataset is properly referenced and aligns with the standards of proper attribution. We appreciate the reviewer’s attention to detail, which has contributed to improving the clarity and accuracy of our work.

---

> > ### Comment · Reviewer_rGQ1 · 2024-11-26
> >
> > I would like to thank the authors for addressing my concerns raised. With that, I have adjusted my scores upwards.

---

### Official Review · Reviewer_Tt6P · 2024-10-31

**Soundness:** 3
**Presentation:** 3
**Contribution:** 3
**Rating:** 8
**Confidence:** 5

**Summary:**

The paper proposes  a novel framework called AVDDCL (Audio-Visual Deception Detection via Cognitive Load). The authors address the challenges of automated deception detection by leveraging cognitive load features extracted from audio-visual data, which eliminates the need for specialized hardware and human annotations.
The key contributions of the paper include:
Novel Framework: The introduction of an automated, scalable deception detection system that incorporates cognitive load features from audio-visual data, enhancing its applicability in real-world scenarios.
Focal Loss Implementation: The use of focal loss to improve the model’s ability to detect difficult instances of deception by emphasizing harder-to-classify cases.
State-of-the-Art Performance: Achieving good results on the DOLOS dataset, demonstrating that integrating cognitive load features also boosts accuracy and robustness in deception detection.

**Strengths:**

1. The paper presents a novel approach to deception detection by integrating cognitive load features from audio-visual data, which sets it apart from traditional methods that rely heavily on physiological signals or manual annotations. This innovation eliminates the need for specialized equipment and subjective inputs, enhancing scalability and practical applicability.
2. The use of focal loss to emphasize harder-to-classify deception cases is a creative adaptation in this context. This approach effectively prioritizes difficult samples, which is critical in the nuanced domain of deception detection
3. The integration of cognitive load into a unified audio-visual framework is an inventive step, providing a more holistic understanding of deception. This combination of existing modalities with cognitive indicators represents a novel application that enhances the accuracy and robustness of deception detection.
4. The paper addresses a problem in deception detection, with potential applications in security, law enforcement, border control, and various other domains. The automated extraction of cognitive load features from audio-visual data can significantly improve the deployment of deception detection systems in real-world settings, where human bias and the requirement for specialized equipment are limiting factors.
5. The model's ability to generalize across both high-stakes and low-stakes deception datasets highlights its versatility, making it a significant contribution to both the research community and industry practitioners focused on automated behavioral analysis.
6. The paper is well-organized, with clear sections detailing the problem, related work, methodology, experiments, and results. The structure allows readers to easily follow the progression of ideas and understand the novelty of the approach.
7. The mathematical formulations are well explained in the paper, aiding in comprehensibility for readers familiar with deep learning frameworks.

**Weaknesses:**

1. Although the paper claims that the model improves scalability by not requiring specialized equipment, the potential limitations in computational cost are not thoroughly addressed. The use of pre-trained models like Wav2Vec2 and Vision Transformers can be computationally expensive, which may hinder real-time deployment in resource-constrained environments. A brief discussion of computational efficiency, inference speed, and potential trade-offs in real-world applications would provide more actionable insights for practitioners.

2. The ethical implications of deploying a system like AVDDCL in real-world settings (e.g., law enforcement, job interviews) are not discussed. The potential for false positives in high-stakes scenarios (e.g., misidentifying truthful individuals as deceptive) could have serious consequences. Addressing these implications, perhaps by suggesting future work in explainability or bias mitigation, would make the contribution more responsible and practical.

**Questions:**

1. Adding details on the selection of the focal loss parameter 𝛾 would improve transparency. Including experiments with different
𝛾 values might also show the sensitivity of AVDDCL to this parameter, thus strengthening the reproducibility of the results.
2. If real-time deployment is a goal, conducting tests or simulations on inference speed and latency would validate the model’s practical usability in settings like live surveillance or border security.

---

> ### Author Response · Authors · 2024-11-20
> **Response to Reviewer Tt6P [1/4]**
>
> Thank you for your valuable feedback and insightful suggestions. We truly appreciate your thorough review and hope that the following responses address your concerns and clarify any uncertainties.
>
> We also welcome any further comments from the reviewer and will make every effort to incorporate them.
>
> **Weakness 1: The Computational Cost Limitations**
>
> We sincerely appreciate the reviewer’s valuable feedback on the computational efficiency and scalability of the AVDDCL framework, particularly in resource-constrained environments. As highlighted in the table below, our AVDDCL model has a lower total parameter count compared to other cognitive load prediction models. This reduction contributes to enhanced computational efficiency, making the model more suitable for deployment in environments with limited resources.
>
> However, we acknowledge that further optimizations are necessary to enable real-time deployment in such environments. To address this, we plan to incorporate techniques such as pruning and quantization, which will help reduce inference latency while preserving performance. These enhancements will ensure that AVDDCL remains practical for real-world applications, particularly in settings with constrained computational resources.
>
> We also acknowledge the importance of balancing computational efficiency with model accuracy for successful and effective deployment. Moving forward, we will continue to explore trade-offs between computational load and performance to ensure that the AVDDCL framework is both efficient and effective across diverse scenarios without compromising accuracy.
>
>
> The table below shows a comparison of cognitive load prediction models based on parameters (Millions) and F1 scores across TLX subscales (M: Mental, E: Effort, T: Temporal).
> | Audio     | Visual         | \# Parameters (Millions) | M    | E    | T    |
> |-----------|----------------|--------------------------|------|------|------|
> | VGG16     | -              | 138.5                    | 58.8 | -    | -    |
> | -         | R(2+1)D-18     | 33.3                     | 60.5 | -    | -    |
> | VGG16     | ResNet3D-18    | 172.1                    | 65.0 | -    | -    |
> | VGG16     | -              | 138.5                    | -    | 58.8 | -    |
> | -         | R(2+1)D-18     | 33.3                     | -    | 65.5 | -    |
> | ResNet18  | R(2+1)D-18     | 43.5                     | -    | 60.8 | -    |
> | ResNet18  | -              | 11.4                     | -    | -    | 58.2 |
> | -         | MC3-18         | 11.6                     | -    | -    | 60.0 |
> | ResNet18  | ResNet3D-18    | 45.4                     | -    | -    | 61.2 |
> | **AVPEF** |                | **71.2 (Trainable 5.2)**  | 63.6 | 61.5 | 58.2 |
>
> The revised and updated details regarding this topic can be found in **Section 4.3**.

---

> ### Author Response · Authors · 2024-11-20
> **Response to Reviewer Tt6P [2/4]**
>
> **Weakness 2: Concerns on the ethical implications of AVDDCL (e.g. the potential for false positives in high-stakes scenarios, and the need for future work in explainability and bias mitigation)**
>
> We sincerely thank the reviewer for emphasizing the critical importance of addressing the ethical implications of deploying AVDDCL in real-world, high-stakes scenarios. We fully agree that issues such as the potential for false positives and their societal consequences must be carefully considered.
>
> To address this, we expanded the **Discussion and Future Work section** to explicitly highlight these challenges. We acknowledge that misidentifying truthful individuals as deceptive in high-stakes contexts, such as law enforcement or job interviews, represents a significant ethical concern. In response, we have outlined several measures to address these ethical implications:
>
> - **Explainability in High-Stakes Scenarios:** While fully automated decision-making systems can improve efficiency, they also raise concerns about fairness and reliability, particularly when human judgment is excluded [1]. In high-stakes scenarios such as law enforcement or medical diagnostics, automation bias can lead to undue reliance on, or dismissal of, system outputs [2]. The potential harm from false positives in these contexts highlights the need for balanced and transparent systems [3]. To mitigate this, we will prioritize improving the explainability of AVDDCL’s predictions. Specifically, we aim to develop interpretable decision-making processes that enable stakeholders to better assess the model’s outputs, thus reducing the risk of over-reliance and ensuring informed and ethical decision-making.
>
> - **Bias Mitigation Strategies:** Incorporating cognitive and psychological factors into system design can enhance trustworthiness [4, 5]. While we utilize cognitive load-based features to address automation bias, unresolved issues such as dataset biases and demographic factors (e.g., gender, age, and culture) remain [6, 7, 8]. These biases limit the model’s generalizability and effectiveness across diverse high- and low-stakes scenarios [9]. To address these challenges, future work will focus on implementing fairness-aware training, demographic balancing, and targeted data augmentation techniques. These strategies aim to promote equitable model performance across diverse populations and minimize the potential for biased outcomes in real-world applications.
>
> - **Human-in-the-Loop Integration:** Incorporating human oversight is crucial to ensuring fairness and accountability alongside automation. Human-in-the-loop systems can complement automated decisions, reducing the risk of misuse or false positives [5, 10]. By integrating human judgment at critical decision points, we can ensure that the model’s outputs are evaluated within the context of real-world complexities. This approach enhances the reliability and accountability of the system and provides an additional safeguard against over-reliance on automated predictions, particularly in high-stakes scenarios where errors could have significant consequences.
>
> We believe these additions significantly strengthen the manuscript and address the reviewer’s concerns comprehensively. Thank you for highlighting this critical area, which has allowed us to refine our work and emphasize the importance of ethical considerations.
>
> The revised and updated details regarding this topic can be found in **Section 5**.

---

> ### Author Response · Authors · 2024-11-20
> **Response to Reviewer Tt6P [3/4]**
>
> **Weakness 2: Concerns on the ethical implications of AVDDCL (e.g. the potential for false positives in high-stakes scenarios, and the need for future work in explainability and bias mitigation)**
>
> [1] Kern, C., Gerdon, F., Bach, R. L., Keusch, F., & Kreuter, F. (2022). Humans versus machines: Who is perceived to decide fairer? Experimental evidence on attitudes toward automated decision-making. Patterns, 3(10).
>
> [2] Belavadi, V., Zhou, Y., Bakdash, J. Z., Kantarcioglu, M., Krawczyk, D. C., Nguyen, L., ... & Thuriasingham, B. (2020, October). MultiModal deception detection: Accuracy, applicability and generalizability. In 2020 Second IEEE International Conference on Trust, Privacy and Security in Intelligent Systems and Applications (TPS-ISA) (pp. 99-106). IEEE.
>
> [3] Biçer, B., & Dibeklioğlu, H. (2023). Automatic Deceit Detection Through Multimodal Analysis of High-Stake Court-Trials. IEEE Transactions on Affective Computing.
>
> [4] Mosier, K. L., & Skitka, L. J. (2018). Human decision makers and automated decision aids: Made for each other?. In Automation and human performance (pp. 201-220). CRC Press.
>
> [5] Cummings, M. L. (2017). Automation bias in intelligent time critical decision support systems. In Decision making in aviation (pp. 289-294). Routledge.
>
> [6] Levitan, S. I., Levitan, Y., An, G., Levine, M., Levitan, R., Rosenberg, A., & Hirschberg, J. (2016, June). Identifying individual differences in gender, ethnicity, and personality from dialogue for deception detection. In Proceedings of the second workshop on computational approaches to deception detection (pp. 40-44).
>
> [7] Naven, G., Sen, T., Gerstner, L., Haut, K., Wen, M., & Hoque, E. (2020, November). Leveraging shared and divergent facial expression behavior between genders in deception detection. In 2020 15th IEEE international conference on automatic face and gesture recognition (FG 2020) (pp. 428-435). IEEE.
>
> [8] Abouelenien, M., Burzo, M., Pérez-Rosas, V., Mihalcea, R., Sun, H., & Zhao, B. (2018). Gender differences in multimodal contact-free deception detection. IEEE MultiMedia, 26(3), 19-30.
>
> [9] King, S. L., & Neal, T. (2024). Applications of AI-Enabled Deception Detection Using Video, Audio, and Physiological Data: A Systematic Review. IEEE Access.
>
> [10] Khan, W., Crockett, K., O'Shea, J., Hussain, A., & Khan, B. M. (2021). Deception in the eyes of deceiver: A computer vision and machine learning based automated deception detection. Expert Systems with Applications, 169, 114341.

---

> ### Author Response · Authors · 2024-11-20
> **Response to Reviewer Tt6P [4/4]**
>
> **Question 1: Concern of the transparency in the selection of the focal loss parameter $\gamma$ and the need for experiments with different $\gamma$ values to improve sensitivity and reproducibility.**
>
> Thank you for raising your concerns regarding using and exploring focal loss in our study. We acknowledge that our initial explanation of the rationale behind the choice of focal loss and its hyperparameter, $\gamma$, was not sufficiently detailed. In response, we conducted extensive tests across a wide range of $\gamma$ values to systematically evaluate their impact on performance. The results of these experiments, encompassing all relevant combinations, are presented in the following table and discussed in detail thereafter.
>
> | Method                  | $\boldsymbol{\gamma}$ | ACC   | F1    | AUC   |
> |-------------------------|-----------------------|-------|-------|-------|
> | AVDDCL (M+E+T)          | 0 (C/E)               | 66.2  | 72.5  | 64.3  |
> | AVDDCL (M+E+T)          | 0.5                   | 66.8  | 72.3  | 65.4  |
> | AVDDCL (M+E+T)          | 1                     | 65.3  | 70.4  | 64.1  |
> | AVDDCL (M+E+T)          | 1.5                   | 66.6  | 72.6  | 64.5  |
> | AVDDCL (M+E+T)          | 2                     | **68.0** | **73.3** | **66.5** |
> | AVDDCL (M+E+T)          | 3                     | 67.5  | 73.3  | 65.7  |
> | AVDDCL (M+E+T)          | 5                     | 66.3  | 70.7  | 65.5  |
>
> To further validate the effectiveness of focal loss, we conducted an ablation study using the DOLOS dataset, testing $\gamma = 0$ (equivalent to cross-entropy loss) alongside a range of other values. The table below summarizes the results of this experiment, demonstrating that focal loss consistently improves model performance compared to cross-entropy loss ($\gamma = 0$). Notably, we observed the highest Accuracy, F1 score, and AUC with $\gamma = 2$, which was subsequently selected as the optimal value for our analyses.
>
> Although focal loss is widely recognized for its ability to address imbalanced data, our findings highlight its specific utility in managing the complexities of distinguishing deceptive instances. Deception often involves subtle cognitive and behavioral indicators, and focal loss enhances the model’s ability to detect and focus on these nuanced patterns. This capability aligns with the inherent challenges of deception detection, where identifying deceptive behavior demands heightened sensitivity to subtle cues.
>
>
> The revised and updated details regarding this topic can be found in **Section 4.6.1**.
>
> **Question 2: Suggestion to test inference speed and latency for real-time deployment in high-stakes environments such as live surveillance or border security.**
>
> We appreciate the reviewer’s suggestion to evaluate inference speed and latency for real-time deployment scenarios. While we primarily focus on validating AVDDCL’s accuracy and generalizability, we fully recognize the importance of optimizing the model for practical use in real-world settings, such as live surveillance or border security.
>
> As part of our future work, we plan to incorporate techniques like model pruning, quantization, and efficient attention mechanisms to reduce inference latency and computational costs. These optimizations will enhance AVDDCL’s scalability and practicality in resource-constrained environments, paving the way for real-time applications.
>
> However, we acknowledge the challenges posed by the absence of diverse real-world datasets, such as the Real-Life Trial, which exacerbates domain-specific overfitting and limits the model’s applicability to scenarios like high- and low-stakes deception detection [1]. The lack of diverse datasets hampers the model’s ability to generalize across different domains effectively. To address this limitation, we aim to prioritize efforts to expand dataset diversity and develop robust domain generalization techniques. These steps are critical to ensuring that AVDDCL can perform reliably across a wide range of real-world scenarios.
>
> Once these foundational challenges are sufficiently addressed, we will focus on optimizing the model for real-time deployment, maximizing its practicality and usability in high-stakes environments.
>
> The revised and updated details regarding this topic can be found in **Section 5**.
>
> [1] King, S. L., & Neal, T. (2024). Applications of AI-Enabled Deception Detection Using Video, Audio, and Physiological Data: A Systematic Review. IEEE Access.

---

### Official Review · Reviewer_xHxN · 2024-11-03

**Soundness:** 2
**Presentation:** 3
**Contribution:** 2
**Rating:** 5
**Confidence:** 4

**Summary:**

This paper presents a multimodal modeling approach for deception detection with a specific focus on measuring cognitive load.  The authors present experiments on several datasets.  The authors propose a feature extraction step (AVPEF) that is specifically focused on extracting cognitive load features. The network is pertaining on the AVCAffe dataset. The results are modest but generally show an improvement over the reported baselines.

**Strengths:**

Overall the paper is fairly clearly written and contains most (but not all) of the necessary details required to replicate the work.  The approach appears to have novelty and the experimental results that are reported (I will discuss below that I think these could be expanded) are reasonable.  The work is motivated and the authors do report some results across three different datasets.

**Weaknesses:**

There are a few areas in which there are important details missing.  For example in the cognitive load feature pre-training section there is only limited details about the pretaining step - which videos were present in the set used (as I understand it some of the videos may no longer be available) how were these preprocessed? Etc.

Why are the evaluation metrics not consistently reported. For example,  in Table 2 accuracy, F1-score and area under the curve are reported but in Table 3 only accuracy is shown (side note: it isn’t actually described in the Table caption or column heading what the metric is - this is a bit sloppy).
Overall, the experiments are a bit fractured, in some cases there are within dataset experiments and in others across dataset experiments.  I think that the experiments could easily be more comprehensive and in their current state it might lead readers to be concerned that there is some cherry-picking of good results going on.

In Section 4.6.1 it isn’t very clear which dataset these results are reported on.  Again, the attention to detail in the paper is not very high and there could be a better description.

Reporting accuracy numbers to four significant figures is a little unnecessary to me as it suggests at we can trust the numbers to that level of precision.  I would limit to 3 significant figures.

Fig 3. Is very hard to reach.  The font is far too small to be legible.  The caption does not describe what projection is used and the reader needs to go to the text to find out that these are t-SNE plots.

There are baselines for the experiments; however, they are quite limited.  I am not sure why the authors do not comment on other performance numbers: e.g. Camara, M. K., Postal, A., Maul, T. H., & Paetzold, G. H. (2024). Can lies be faked? Comparing low-stakes and high-stakes deception video datasets from a Machine Learning perspective. Expert Systems with Applications, 249, 123684.
If the reason for this is that the authors do not do cross-fold/within subjects experiments on that dataset then I don’t think that is a good reason, it would be better if they were to match those experiments rather than exclude them.

**Questions:**

See above.

---

> ### Author Response · Authors · 2024-11-20
> **Response to Reviewer xHxN [1/4]**
>
> Thank you for your valuable feedback and insightful suggestions. We truly appreciate your thorough review and hope that the following responses address your concerns and clarify any uncertainties.
>
> We also welcome any further comments from the reviewer and will make every effort to incorporate them.
>
> **Weaknesses 1: Further clarification is needed on cognitive load feature pre-training, such as the videos used and preprocessing steps.**
>
> We sincerely appreciate the reviewer’s observation and acknowledge the need for greater transparency and detail in the cognitive load feature pre-training section.
>
> To address this, we presented additional details specifying that the pre-training dataset, provided by AVCAffe, contains 58,112 short video segments, each averaging 6.74 seconds in duration, totaling approximately 108.72 hours of content from 106 participants.
>
> Furthermore, we add a **data pre-processing and statistics** section to detail the preprocessing steps for audio and image data, ensuring greater clarity and transparency. A consistent preprocessing pipeline was applied across all datasets, including AVCAffe, DOLOS, RLT, and BOL. Specifically, 64 frames were uniformly selected from each video, and facial regions were isolated using the MTCNN face detector [1]. These face images were resized to 160x160 pixels and normalized. For audio, speech signals were resampled to produce 64 tokens suitable for the W2V2 feature extractor, and the Demucs model was utilized to separate speech from background noise [2].
>
> We are grateful to the reviewer for identifying this area for improvement, which has allowed us to provide a more detailed and transparent account of our methodology.
>
> The revised and updated details regarding this topic can be found in **Appendix A.4**.
>
> **Weakness 2: Inconsistent reporting of evaluation metrics across tables, with opportunities to clarify metric descriptions in captions and headings.**
>
> We sincerely thank the reviewer for pointing out the inconsistency in the reported evaluation metrics. To address this, we have included F1-score and AUC for all experiments across all relevant tables in the main manuscript, ensuring consistency in the evaluation metrics presented.
>
> This strategy enables us to maintain clarity in the main manuscript while ensuring that our results are presented with full transparency and detail. Additionally, we have revised the captions and column headings to explicitly indicate the metrics being reported, directly addressing the reviewer's concern and enhancing the overall quality of the presentation.
>
>
> [1] Zhang, K., Zhang, Z., Li, Z., & Qiao, Y. (2016). Joint face detection and alignment using multitask cascaded convolutional networks. IEEE signal processing letters, 23(10), 1499-1503.
>
> [2] Défossez, A., Usunier, N., Bottou, L., & Bach, F. (2019). Music source separation in the waveform domain. arXiv preprint arXiv:1911.13254.

---

> ### Author Response · Authors · 2024-11-20
> **Response to Reviewer xHxN [2/4]**
>
> **Weakness 3: The experiments vary between within-dataset and across-dataset settings, which could benefit from greater consistency to provide a more comprehensive evaluation**
>
> **Weakness 7: Limited baselines and lack of comparison with key prior work, such as Camara et al. (2024), which raises concerns about the exclusion of relevant experiments.**
>
> We sincerely appreciate the reviewer’s feedback regarding the experimental design, baseline comparisons, and the concerns related to consistency and broader evaluations.
>
> In response, we completely restructured our experimental approach to ensure a more systematic and comprehensive evaluation of AVDDCL. To achieve a fair and rigorous comparison, we implemented a 5-fold cross-validation strategy across both the RLT and BOL datasets. For the BOL dataset, we adjusted the initial utterance-based labels by grouping them into rounds, aligning with the video-based labeling format utilized in the RLT dataset. Additionally, we addressed the class imbalance inherent in the BOL dataset by applying random under-sampling to the 'deception' class, resulting in a balanced dataset that enabled the model to effectively learn both deception and truth categories without bias. Consequently, the RLT dataset consisted of 111 videos, while the balanced BOL dataset comprised 128 videos. Detailed dataset statistics are included in the Appendix.
>
> The redesigned experiments included both within-dataset and cross-corpus evaluations to thoroughly assess the adaptability and generalizability of AVDDCL. The results, presented in the table below, demonstrate consistent and robust performance across a variety of scenarios. Metrics such as ACC, F1-score, and AUC were utilized to ensure a comprehensive evaluation.
>
> We would like to highlight that all experimental outcomes, including both strengths and limitations, were reported with full transparency. The redesigned experiments were meticulously structured to provide a fair and well-rounded assessment of the proposed framework.
>
> The revised and updated details regarding this topic can be found in **Section 4.5** and **Appendix A.4**.
>
> | **Model**            | **Modality** | **Train RLT / Test RLT**       |                        |             | **Train BOL / Test BOL**       |                        |             |
> |----------------------|--------------|--------------------------------|------------------------|-------------|--------------------------------|------------------------|-------------|
> |                      |              | **ACC**                       | **F1-score**           | **AUC**     | **ACC**                       | **F1-score**           | **AUC**     |
> | Camara et al. (2024) [1] | V            | 62.7 ± 1.0                         | 62.8 ± 1.0                | 64.1 ± 1.1        | 62.4  ± 1.2                         | 58.2  ± 2.2                 | 62.4  ± 1.2       |
> | Guo et al. (2023) [3]  | V + A        | 82.7 ± 1.3                         | 83.3 ± 1.2                 | 82.8 ± 1.3        | 67.0  ± 1.6                         | 62.1  ± 2.4                 | 67.0  ± 1.2       |
> | **AVDDCL (Ours)**    | V + A        | **86.4 ± 1.4**                      | **85.6 ± 1.6**              | **86.6 ± 1.4**    | **74.0 ± 1.9**                      | **68.5 ± 3.5**              | **74.0 ± 1.9**    |
>
> Table: Within-dataset experiments on RLT (high-stakes) and BOL (low-stakes) datasets.
>
> ---
>
> | **Model**            | **Modality** | **Train RLT / Test BOL**       |                        |             | **Train BOL / Test RLT**       |                        |             |
> |----------------------|--------------|--------------------------------|------------------------|-------------|--------------------------------|------------------------|-------------|
> |                      |              | **ACC**                       | **F1-score**           | **AUC**     | **ACC**                       | **F1-score**           | **AUC**     |
> | Camara et al. (2024) [1] | V            | 42.8                          | 39.8                  | 50.9        | 50.2                          | 46.6                  | 49.7        |
> | Biccer et al. (2023) [2] | V            | -                             | 44.1                  | -           | -                             | 44.7                  | -           |
> | Biccer et al. (2023) [2] | A            | -                             | 38.2                  | -           | -                             | 45.6                  | -           |
> | Guo et al. (2023) [3]   | V + A        | 48.6                          | 15.7                  | 48.6        | 52.6                          | 58.8                  | 53.2        |
> | **AVDDCL (Ours)**    | V + A        | 47.4                          | 7.1                   | 47.3        | **55.3**                      | 57.6                  | **55.5**    |
>
> Table: Cross-corpus experiments on RLT (high-stakes) and BOL (low-stakes) datasets.

---

> ### Author Response · Authors · 2024-11-20
> **Response to Reviewer xHxN [4/5]**
>
> **Weakness 3: The experiments vary between within-dataset and across-dataset settings, which could benefit from greater consistency to provide a more comprehensive evaluation**
>
> **Weakness 7: Limited baselines and lack of comparison with key prior work, such as Camara et al. (2024), which raises concerns about the exclusion of relevant experiments.**
>
> As shown in the above tables in Response to Reviewer xHxN [2/5], our AVDDCL model demonstrates strong performance in within-dataset experiments, achieving an accuracy of 86.4% on high-stakes (RLT) deception and 74.0% on low-stakes (BOL) deception. These results reflect the model's ability to effectively capture domain-specific patterns.
>
> In the cross-dataset experiments, where the model is trained on one dataset and tested on another, performance varies. When trained on BOL and tested on RLT, the accuracy is 55.3%, indicating that certain deception patterns are transferable across datasets. Conversely, when trained on RLT and tested on BOL, the accuracy drops to 47.4%, presenting that high-stakes deception, characterized by more pronounced cues, is relatively easier for the model to detect [4]. In contrast, low-stakes deception involves subtler cues, making generalization more challenging for the model.
>
> Moreover, while the model performs well overall, the F1 scores in the Train RLT / Test BOL setting are lower than in the Train BOL / Test RLT setting. This disparity highlights reduced classification accuracy across domains, particularly in low-stakes contexts.
>
> These findings highlight the challenges associated with domain differences in deception detection. While AVDDCL excels in within-dataset settings, incorporating cognitive load features significantly enhances its performance and improves generalization from low-stakes to high-stakes scenarios. However, further refinement is required to increase the model's sensitivity to the subtle cues associated with low-stakes deception. Future research should aim to enhance the model's ability to capture these nuances, thereby improving cross-domain performance and strengthening its generalization capabilities across diverse deception contexts.
>
> [1] Camara, M. K., Postal, A., Maul, T. H., & Paetzold, G. H. (2024). Can lies be faked? Comparing low-stakes and high-stakes deception video datasets from a Machine Learning perspective. Expert Systems with Applications, 249, 123684.
>
> [2] Biçer, B., & Dibeklioğlu, H. (2023). Automatic Deceit Detection Through Multimodal Analysis of High-Stake Court-Trials. IEEE Transactions on Affective Computing.
>
> [3] Guo, X., Selvaraj, N. M., Yu, Z., Kong, A. W. K., Shen, B., & Kot, A. (2023). Audio-visual deception detection: Dolos dataset and parameter-efficient crossmodal learning. In Proceedings of the IEEE/CVF International Conference on Computer Vision (pp. 22135-22145).
>
> [4] Wright Whelan, C., Wagstaff, G. F., & Wheatcroft, J. M. (2014). High-stakes lies: Verbal and nonverbal cues to deception in public appeals for help with missing or murdered relatives. Psychiatry, Psychology and Law, 21(4), 523-537.

---

> ### Author Response · Authors · 2024-11-20
> **Response to Reviewer xHxN [4/4]**
>
> **Weakness 4: Concern about clarity regarding the dataset, and the need for more detailed descriptions throughout the paper**
>
> We have revised the manuscript to improve the clarity and precision of the dataset descriptions and metrics reported throughout the paper. Adjustments were examined to ensure that the results were presented unambiguously, facilitating a clearer understanding of the datasets and the evaluation metrics employed.
>
> We sincerely thank the reviewer for bringing this to our attention and for providing valuable feedback that allowed us to enhance the manuscript.
>
> **Weakness 5: Concern about reporting accuracy to four significant figures, suggesting unnecessary precision. It is recommended to limit to 3 significant figures.**
>
> We sincerely appreciate the reviewer’s comment regarding the level of precision in reporting accuracy metrics. In response, we have revised the manuscript to consistently report accuracy with three significant figures.
>
> This adjustment strikes an appropriate balance, ensuring precision while avoiding overestimation of the results' reliability. Thank you for highlighting this point, as we believe this change enhances the clarity and consistency of the reported metrics.
>
> **Weakness 6: Fig. 3 is hard to interpret due to the small font size, and the caption lacks information about the t-SNE projection.**
>
> The font size in Fig. 3 is increased, and the caption specifies the use of t-SNE projection, enhancing readability and clarity.
>
> Moreover, we have taken steps to improve the captions throughout the manuscript to ensure consistency and better comprehension. We sincerely thank the reviewer for their valuable feedback, which helped us make these improvements.

---

> > ### Comment · Reviewer_xHxN · 2024-11-25
> > **Response to rebuttal**
> >
> > I would like to thank the authors for their rebuttal.  Overall, my main comments have been addressed.  There are still a few things that would help improve the paper.
> >
> > - After running 5 fold cross validation it is very helpful to show the range of performance metrics across folds in addition to the mean.
> >
> > - The axes for Fig.3 are still unreadable. If the numbers on the x and y axes are not meaningful just remove them, if they are important then they need to be bigger.  My guess is that the former is probably the case.
> >
> > - I would argue there is still not enough information about the preprocessing/pretraining for someone to fully replicate the work.
> > "For pre-training, we utilize the AVCAffe dataset (Sarkar et al., 2023),
> > which includes 58,112 short video segments averaging 6.74 seconds each, totaling approximately
> > 108.72 hours of data from 106 participants."
> >
> > Any comments on resolution of the frames/if they were resized, what resizing function was used, etc. There are a number of these types of details that I couldn't find in the paper.
> >
> > If the authors could make a couple of further improvements I would be filling to increase my score.

---

> ### Author Response · Authors · 2024-11-26
> **Additional Response (1/2)**
>
> **Question 1: After running 5-fold cross-validation it is very helpful to show the range of performance metrics across folds in addition to the mean.**
>
> We sincerely thank the reviewer for this valuable suggestion. In response to your feedback, we revised the table to include the mean and its variation for each metric (Accuracy, F1-score, and AUC) over the 5-fold cross-validation.
>
> This addition provides a clearer and more comprehensive representation of the variability across folds, giving readers a better understanding of the model’s consistency. By incorporating some variations, we aim to deliver a more transparent and robust evaluation of the proposed framework.
>
> We greatly appreciate your recommendation, as it allows us to enhance the clarity and informativeness of our results. The revised table now offers deeper insights into the stability and reliability of our approach across different data splits.
>
> | **Model**            | **Modality** | **Train RLT / Test RLT**       |                        |             | **Train BOL / Test BOL**       |                        |             |
> |----------------------|--------------|--------------------------------|------------------------|-------------|--------------------------------|------------------------|-------------|
> |                      |              | **ACC**                       | **F1-score**           | **AUC**     | **ACC**                       | **F1-score**           | **AUC**     |
> | Camara et al. (2024) | V            | 62.7 ± 1.0                         | 62.8 ± 1.0                | 64.1 ± 1.1        | 62.4  ± 1.2                         | 58.2  ± 2.2                 | 62.4  ± 1.2       |
> | Guo et al. (2023)   | V + A        | 82.7 ± 1.3                         | 83.3 ± 1.2                 | 82.8 ± 1.3        | 67.0  ± 1.6                         | 62.1  ± 2.4                 | 67.0  ± 1.2       |
> | **AVDDCL (Ours)**    | V + A        | **86.4 ± 1.4**                      | **85.6 ± 1.6**              | **86.6 ± 1.4**    | **74.0 ± 1.9**                      | **68.5 ± 3.5**              | **74.0 ± 1.9**    |
>
> --
>
> **Question 2: The axes for Fig.3 are still unreadable. If the numbers on the x and y axes are not meaningful just remove them, if they are important then they need to be bigger. My guess is that the former is probably the case.**
>
> Thank you for highlighting the issue with the readability of the axes in Fig. 3. We appreciate your observation and have carefully reviewed the visualization.
>
> Following your suggestion, we removed the numerical labels from the *x* and *y* axes, as they were not essential for interpreting the figure. Moreover, we made several adjustments to the overall design of the visualization to improve clarity and readability.
>
> We are grateful to the reviewer for bringing this to our attention, as it allows us to enhance the quality and accessibility of our figure. The revised Fig. 3 is now included in the updated manuscript.

---

> ### Author Response · Authors · 2024-11-26
> **Additional Response (2/2)**
>
> **Question 3: I would argue there is still not enough information about the preprocessing/pretraining for someone to fully replicate the work.**
>
> ***For pre-training, we utilize the AVCAffe dataset (Sarkar et al., 2023), which includes 58,112 short video segments averaging 6.74 seconds each, totaling approximately 108.72 hours of data from 106 participants***
>
> **Any comments on the resolution of the frames/if they were resized, what resizing function was used, etc. There are a number of these types of details that I couldn't find in the paper.**
>
> We sincerely thank the reviewer for this valuable suggestion on the applicability of our paper.
>
> **Video Pre-Processing**: The video preprocessing pipeline was designed to extract meaningful facial regions from video frames while maintaining temporal consistency. The steps involved are as follows:
>
> - Frames were uniformly sampled at a fixed rate of 20 frames per second (FPS) using OpenCV. For videos with FPS greater than 20, frames were sampled at intervals calculated as *frame-interval* = max(1, ⌊fps/20⌋). For videos with FPS less than 20, no frames were skipped.
> - Facial regions were detected using the MTCNN face detector, which provided bounding box coordinates and keypoints for facial landmarks (e.g., eyes, nose, mouth). In cases where multiple faces were detected, a tracking mechanism was implemented to assign unique IDs based on bounding box positions and an Euclidean distance threshold of 40 pixels between nose keypoints. The most frequently detected face was selected as the primary face.
> - The identified facial regions were cropped and resized to a fixed resolution of 160$\times$160 pixels using OpenCV's *cv2.resize()* function with bilinear interpolation to preserve facial details.
> - A total of 64 frames were uniformly sampled per video to ensure consistent representation across datasets.
>
> **Audio Pre-Processing**: The audio preprocessing pipeline was designed to ensure clean and consistent speech signals while minimizing background noise. The steps involved are as follows:
>
> - Audio tracks were extracted from each video using the MoviePy library's *VideoFileClip* class. The extracted audio was saved as *.wav* files.
> - Videos without audio tracks were logged and excluded from further processing to maintain pipeline integrity.
> - Speech signals were refined using the advanced Demucs model [1, 2], a deep learning-based source separation model. The two-stem configuration was used to isolate vocals (speech) from background noise.
> - The refined speech signals were resampled to ensure compatibility with the Wav2Vec 2.0 (W2V2) feature extractor, producing 64 tokens per audio clip.
>
> These detailed steps ensure reproducibility and consistency in preprocessing across all datasets, including AVCAffe, DOLOS, RLT, and BOL.
>
> **Training Details for AVCAffe**: For cognitive load feature extraction, experiments were conducted using the AVCAffe dataset. The data-splitting strategy described in Section 3.2.1 was applied to ensure no information leakage between training and validation sets. The training details are as follows:
>
> - The model was trained with a learning rate of $3 \times 10^{-4}$ and a batch size of 8.
> - Cross-entropy loss was used as the objective function.
> - The architecture included four encoders, with dropout applied for regularization.
> - Training was conducted for 20 epochs using the Adam optimizer.
> - An early stopping mechanism based on F1-score was employed, halting training if no improvement was observed for 5 consecutive epochs.
>
> These steps are thoroughly documented in **Section 4.3**, with further implementation details available in **Appendix A.5** to facilitate reproducibility. We hope this addresses the reviewer's concerns and enhances the clarity of the manuscript.
>
>
> [1] Jade Copet, Felix Kreuk, Itai Gat, Tal Remez, David Kant, Gabriel Synnaeve, Yossi Adi, and Alexan- dre D´efossez. Simple and controllable music generation. *Advances in Neural Information Processing Systems, 36*, 2024.
>
> [2] Alexandre D´efossez, Nicolas Usunier, L´eon Bottou, and Francis Bach. Music source separation in
> the waveform domain. arXiv preprint arXiv:1911.13254, 2019.

---

> > ### Comment · Reviewer_xHxN · 2024-12-03
> > **Response to Rebuttal**
> >
> > My review and score have been updated to reflect the responses. Thank you.

---

> > > ### Author Response · Authors · 2024-12-03
> > >
> > > Thank you for your valuable comments and consideration on our manuscript. We would like to express our sincere gratitude to the reviewers. We are happy to have a great chance to improve the manuscript!

---

### Official Review · Reviewer_6r1s · 2024-11-04

**Soundness:** 3
**Presentation:** 3
**Contribution:** 3
**Rating:** 8
**Confidence:** 4

**Summary:**

The paper explores the task of deception detection using audio visual modalities. The authors propose that deception requires significantly different cognitive strain or load and hence nudging a prediction network in that direction can lead to a better performance. The authors describe an architecture based on recent feature extractors for a similar task and conduct various experiments to describe the effectiveness of their approach.

**Strengths:**

1. The paper proposes the use of cognitive load or TLX-based features as an intermediary subtask and the framework in itself and the idea is quite novel
2. The paper is well structured and the core architecture is easy to understand
3. The authors provide a lot of details about their implementation to help reporducibility

**Weaknesses:**

While the paper aims at solving a novel problem and definitely brings in a unique perspective, here are some aspects which need to be answered further:
1. If focal loss is shown as a key contribution, it is not well explored in writing and experimentation. Focal loss is widely popular and is often among the standard practices for handling imbalanced data. Claims of introducing it as a new loss function might need to be softened. Even the experiments pre-set the loss param at 2 and no justification or experiments are shown towards why that was chosen or insights on the use of the loss function is missing.
2. Related work doesn't describe the gaps effectively. For example line 137, "... no existing research has explored the automatic extraction of cognitive load indicators through multimodal features." This is not true since AVCaffe (the dataset used) itself does this. Also it doesn't point to the gap that the current work is trying to address which is more towards the use of cognitive load to inspire deception detection
3. The motivation behind using AVPEF as feature extractors for TLX states' estimation is unclear. While it worked well for the overall task of deception detection why might it do well for detecting cognitive load states?
4. Given that the dataset is imbalanced the authors should clarify if all metrics are balanced for the same (e.g. macro F1, balanced accuracy) or justify their choice of metric
5. Since related work does not directly quantify how cognitive load and deception are related, showing correlations/associations even between predicted values of TLX subscales and deception as a small experiment will help strengthen the impact of the framework for future researchers

**Questions:**

1. Why as AVPEF chosen to be the feature extractor? How does it compare against other popular feature extraction models?
2. Why was all the training/analysis done on DOLOS when RLT and BOL datasets also exist? They only appear in the draft under Section 4.5 and given how Guo et. al. performs on generalization of the two, they might be very differently distributed and incorporating that in training might in fact make the model more robust.

---

> ### Author Response · Authors · 2024-11-20
> **Response to Reviewer 6r1s [1/5]**
>
> Thank you for your valuable feedback and insightful suggestions. We truly appreciate your thorough review and hope that the following responses address your concerns and clarify any uncertainties.
>
> We also welcome any further comments from the reviewer and will make every effort to incorporate them.
>
> **Weakness 1: Insufficient explanation of parameter selection and experimentation for focal loss, limiting its justification as a key contribution.**
>
> Thank you for bringing up your concerns regarding using and exploring focal loss in our study. We acknowledge that our initial explanation of the choice of focal loss and its hyperparameter, $\gamma$, lacked the necessary depth. In response, we conducted comprehensive tests across a broad range of $\gamma$ values to systematically assess the impact on performance. The results of these tests, covering all relevant combinations, are presented in the following table and discussed in detail.
> The metrics are ACC (%), F1 (%), and AUC (%). Note that C/E represents Cross Entropy Loss.
>
> | Method                  | $\boldsymbol{\gamma}$ | ACC   | F1    | AUC   |
> |-------------------------|-----------------------|-------|-------|-------|
> | AVDDCL (M+E+T)          | 0 (C/E)               | 66.2  | 72.5  | 64.3  |
> | AVDDCL (M+E+T)          | 0.5                   | 66.8  | 72.3  | 65.4  |
> | AVDDCL (M+E+T)          | 1                     | 65.3  | 70.4  | 64.1  |
> | AVDDCL (M+E+T)          | 1.5                   | 66.6  | 72.6  | 64.5  |
> | AVDDCL (M+E+T)          | 2                     | **68.0** | **73.3** | **66.5** |
> | AVDDCL (M+E+T)          | 3                     | 67.5  | 73.3  | 65.7  |
> | AVDDCL (M+E+T)          | 5                     | 66.3  | 70.7  | 65.5  |
>
> To further validate the effectiveness of focal loss, we performed an ablation study using the DOLOS dataset, experimenting with $\gamma = 0$ (equivalent to cross-entropy loss) and a variety of other values. The table below provides a summary of this experiment, which illustrates that focal loss consistently enhances model performance in comparison to cross-entropy loss ($\gamma = 0$). Specifically, we achieved the highest Accuracy, F1 score, and AUC with $\gamma = 2$, which we selected as the optimal value for our subsequent analyses.
>
> While focal loss is well-known for its ability to handle imbalanced data, our findings demonstrate its particular effectiveness in addressing the complexities of distinguishing deceptive instances, which often involve subtle cognitive and behavioral indicators. This finding aligns with the unique challenges of deception detection, where identifying deceptive behavior requires models to detect and focus on nuanced patterns.
>
> The revised and updated details regarding this topic can be found in **Section 4.6.1**.
>
> **Weakness 2: The related work section lacks clarity on research gaps and distinctions from AVCAffe.**
>
> We sincerely apologize for any confusion that may be presented by our initial phrasing, and we are grateful to the reviewer for their feedback on clarifying the research gaps and distinguishing our work from prior research, including the AVCaffe dataset.
>
> While the AVCaffe dataset is presented as a valuable benchmark for analyzing cognitive load through audio-visual data, it does not specifically target deception detection. Our work builds upon this foundation by introducing the first approach that automatically extracts cognitive load indicators specifically for deception detection tasks.
>
> This distinction emphasizes the novelty of our approach, as we utilize cognitive load features directly as predictive signals for identifying deception. By addressing this critical gap—bridging cognitive load analysis with deception detection—we contribute to the expansion of multimodal analysis for practical applications. In addition, we would like to highlight that our approach is both scalable and automated, leveraging cognitive load features to tackle real-world challenges in deception detection.
>
> The revised and updated details regarding this topic can be found in **Section 2.2**.

---

> ### Author Response · Authors · 2024-11-20
> **Response to Reviewer 6r1s [2/5]**
>
> **Weakness 3 & Question 1: Concerns about the choice of AVPEF for TLX state estimation, with limited justification and comparison to other feature extractors.**
>
> Thank you for highlighting the need to provide greater clarity on our motivation for selecting AVPEF as the feature extractor for TLX state estimation. We outline the key reasons supporting our choice as follows:
>
> - **Baseline Comparisons in AVCAffe**: We present a detailed comparison of AVPEF's performance on the AVCAffe dataset [1] against a representative approach outlined. In their study, various combinations of backbones were explored: for the audio modality, they used combinations of VGG and ResNet, while for the visual modality, architectures such as ResNet3D, R(2+1)D, and MC3 were examined. The table below highlights the best-performing backbone combinations for each TLX subscale (Mental Demand, Effort, Temporal Demand) based on F1 scores, and also provides a comparison of total and trainable parameters, offering a comprehensive analysis of model efficiency.
>
> - **Parameter Efficiency**: AVPEF demonstrates exceptional parameter efficiency, utilizing only 5.2M trainable parameters—significantly fewer than many other multimodal or unimodal feature extractors, which often require tens of millions of parameters. Unlike other models in the table, which show identical values for total and trainable parameters, AVPEF's reduced parameter count decreases computational costs and enhances efficiency, particularly in resource-constrained environments. By leveraging pre-trained ViT and W2V2 models alongside a uniform temporal adapter, AVPEF achieves effective learning across diverse settings while minimizing computational demands.
>
> - **Importance of Temporal Information**: Cognitive load is intrinsically linked to temporal dynamics, as illustrated in several studies [2,3]. AVPEF, equipped with a uniform temporal encoder, excels at capturing temporal variations in cognitive load states. This capability allows AVPEF to accurately measure cognitive load, a task where existing baseline models may overlook critical temporal dependencies.
>
> - **Baseline Inconsistencies**: Our analysis indicates that baseline models exhibit inconsistent performance across different modalities and TLX subscales. The optimal choice of feature extractors varies for each TLX subscale, leading to experimental complexity and reduced generalizability. In contrast, AVPEF offers stable and reliable performance across both modalities and subscales, ensuring consistency in its predictions.
>
> - **Comparable Performance**: Despite its lower parameter count, AVPEF achieves F1 scores that are comparable to other models for TLX subscales, including Mental Demand, Effort, and Temporal Demand. This combination of efficiency and accuracy positions AVPEF as a practical and reliable option for cognitive load prediction.
>
> The table below shows a comparison of cognitive load prediction models based on parameters (Millions) and F1 scores across TLX subscales (M: Mental, E: Effort, T: Temporal).
> | Audio     | Visual         | \# Parameters (Millions) | M    | E    | T    |
> |-----------|----------------|--------------------------|------|------|------|
> | VGG16     | -              | 138.5                    | 58.8 | -    | -    |
> | -         | R(2+1)D-18     | 33.3                     | 60.5 | -    | -    |
> | VGG16     | ResNet3D-18    | 172.1                    | 65.0 | -    | -    |
> | VGG16     | -              | 138.5                    | -    | 58.8 | -    |
> | -         | R(2+1)D-18     | 33.3                     | -    | 65.5 | -    |
> | ResNet18  | R(2+1)D-18     | 43.5                     | -    | 60.8 | -    |
> | ResNet18  | -              | 11.4                     | -    | -    | 58.2 |
> | -         | MC3-18         | 11.6                     | -    | -    | 60.0 |
> | ResNet18  | ResNet3D-18    | 45.4                     | -    | -    | 61.2 |
> | **AVPEF** |                | **71.2 (Trainable 5.2)**  | 63.6 | 61.5 | 58.2 |
>
> This comprehensive analysis demonstrates AVPEF's effectiveness in cognitive load prediction and its potential for scalable application in real-world scenarios, prioritizing computational efficiency and robust performance.
>
> The revised and updated details regarding this topic can be found in **Section 4.3**
>
> [1] Sarkar, P., Posen, A., & Etemad, A. (2023, June). AVCAffe: a large scale audio-visual dataset of cognitive load and affect for remote work. In Proceedings of the AAAI Conference on Artificial Intelligence (Vol. 37, No. 1, pp. 76-85).
>
> [2] Puma, S., Matton, N., Paubel, P. V., & Tricot, A. (2018). Cognitive load theory and time considerations: Using the time-based resource sharing model. Educational Psychology Review, 30, 1199-1214.
>
> [3] Liu, Y., Yu, Y., Ye, Z., Li, M., Zhang, Y., Zhou, Z., ... & Zeng, L. L. (2023). Fusion of spatial, temporal, and spectral EEG signatures improves multilevel cognitive load prediction. IEEE Transactions on Human-Machine Systems, 53(2), 357-366.

---

> ### Author Response · Authors · 2024-11-20
> **Response to Reviewer 6r1s [3/5]**
>
> **Weakness 4: Concerns regarding how metrics address dataset imbalance and the rationale behind metric selection.**
>
> For the DOLOS dataset, the Deception/Truth ratio is 1.16, suggesting that the dataset is relatively balanced between deceptive and truthful samples. Similarly, the Real Life Trial dataset has a Deception/Truth ratio of 0.92, while the Box of Lies dataset maintains an ideal balance with a ratio of 1.0.
>
> Given the balanced nature of these datasets, we employed standard metrics such as Accuracy and F1-score to assess the model's performance. These metrics are well-suited for datasets with minimal class imbalance, as they offer fair and interpretable evaluations without bias towards any particular class. This consistent evaluation strategy ensures an accurate reflection of the model's performance across all datasets.
>
> The table below provides a summary of the datasets and evaluation metrics used by the models cited in our study:
>
> | Model                         | Dataset                   | Metric                                      |
> |-------------------------------|---------------------------|---------------------------------------------|
> | Camara et al. (2024) [1]      | RLT, BOL                  | ACC                                         |
> | Guo et al. (2023) [2]         | DOLOS                     | ACC, F1, AUC                                |
> | Chebbi & Jebara (2023) [3]    | RLT                       | ACC, F1                                     |
> | Hossain Sakib et al. (2024) [4]| RLT                       | ACC, Precision, Recall, AUC                |
> | Li et al. (2024) [5]          | RLT, BOL, Bag of Lies      | ACC, F1                                     |
> | **AVDDCL (Ours)**             | DOLOS, RLT, BOL           | ACC, F1, AUC                                |
>
>
> [1] Camara, M. K., Postal, A., Maul, T. H., & Paetzold, G. H. (2024). Can lies be faked? Comparing low-stakes and high-stakes deception video datasets from a Machine Learning perspective. Expert Systems with Applications, 249, 123684.
>
> [2] Guo, X., Selvaraj, N. M., Yu, Z., Kong, A. W. K., Shen, B., & Kot, A. (2023). Audio-visual deception detection: Dolos dataset and parameter-efficient crossmodal learning. In Proceedings of the IEEE/CVF International Conference on Computer Vision (pp. 22135-22145).
>
> [3] Chebbi, S., & Jebara, S. B. (2023). Deception detection using multimodal fusion approaches. Multimedia Tools and Applications, 82(9), 13073-13102.
>
> [4] Hossain Sakib, M. K., Islam, M. R., Akter Prome, S., Nguyen, T. T. L., Asirvatham, D., Ari Ragavan, N., ... & Sanin, C. (2024, April). MVis4LD: Multimodal visual interactive system for lie detection. In Asian Conference on Intelligent Information and Database Systems (pp. 28-43). Singapore: Springer Nature Singapore.
>
> [5] Li, Z., Yu, Z., Lin, X., Selvaraj, N. M., Guo, X., Shen, B., ... & Kot, A. (2024, September). Flexible-modal deception detection with audio-visual adapter. In 2024 IEEE International Joint Conference on Biometrics (IJCB) (pp. 1-10). IEEE.

---

> ### Author Response · Authors · 2024-11-20
> **Response to Reviewer 6r1s [4/5]**
>
> **Weakness 5: Suggestion to include quantitative analysis of the relationship between cognitive load and deception to strengthen the framework's impact for future researchers.**
>
> We sincerely thank the reviewer for their insightful suggestion to explore the correlations between TLX subscales and deception labels. The connection between cognitive load dimensions and deception is a relatively underexplored topic, particularly within the context of deception detection.
>
> To address this, we utilized the pre-trained AVPEF network as described in **Section 3.2.1**, retaining the linear layers. This configuration allowed us to categorize each cognitive load subscale (Mental Demand, Effort, and Temporal Demand) into high or low levels, which were then compared with deception labels in the DOLOS dataset.
>
>
>
> | Cognitive Load Subscale    | Deception/Truth | Low  | High |
> |----------------------------|-----------------|------|------|
> | **Mental Demand**           | Deception       | 77   | 811  |
> |                            | Truth           | 71   | 697  |
> | **Effort**                  | Deception       | 547  | 341  |
> |                            | Truth           | 448  | 320  |
> | **Temporal Demand**         | Deception       | 156  | 732  |
> |                            | Truth           | 147  | 621  |
>
>
> The table above presents the confusion matrix results, showing the distribution of deception and truth labels across high and low levels for each cognitive load subscale. This analysis aims to provide an initial insight into the interactions between cognitive load dimensions and deception.
>
> Our findings are summarized as follows:
>
> - **Correlation Analysis**: We employed confusion matrices and statistical tests, including Cramér’s V, to examine the associations between TLX subscale classifications (high/low) and deception labels. The results indicated that the individual subscales demonstrated weak correlations with deception labels, with Cramér’s V values below 0.1 across all dimensions. This suggests that, when considered independently, these subscales may not significantly impact deception classification. These findings lay the groundwork for investigating more robust or combined features in future research.
>
> - **Combined Insights**: While individual subscales showed limited correlation, prior research [1] and our results suggest that a combination of multiple cognitive load dimensions may more accurately capture the mental states associated with deception. This aligns with our framework’s approach of integrating cognitive load features to improve detection accuracy.
>
> - **Future Implications**: These findings provide preliminary evidence of a complex relationship between cognitive load and deception. We agree that further investigation into these associations can enhance the framework’s effectiveness. In future work, we intend to conduct larger-scale studies incorporating additional datasets and more advanced statistical models to better quantify these relationships.
>
> By addressing this feedback, we believe the manuscript now offers a more in-depth discussion of cognitive load’s relevance to deception detection and its potential implications for future research. We sincerely thank the reviewer for bringing attention to this important area of inquiry.
>
> The revised and updated details regarding this topic can be found in **Appendix A.6**.
>
> [1] Nikulin, C., Lopez, G., Piñonez, E., Gonzalez, L., & Zapata, P. (2019). NASA-TLX for predictability and measurability of instructional design models: case study in design methods. Educational Technology Research and Development, 67, 467-493.

---

> ### Author Response · Authors · 2024-11-20
> **Response to Reviewer 6r1s [5/5]**
>
> **Question 2: Suggestion to incorporate within-dataset experiments on RLT and BOL.**
>
> Thank you for pointing out the limitation of focusing exclusively on the DOLOS dataset and for emphasizing the potential benefits of incorporating the RLT and BOL datasets. Your feedback has allowed us to broaden our experiments and provide a more comprehensive evaluation of the proposed AVDDCL framework.
>
> In response, we conducted additional experiments by training and evaluating AVDDCL on both the RLT and BOL datasets. These datasets represent high-stakes (RLT) and low-stakes (BOL) deception scenarios, offering a wider context to thoroughly assess the model's performance.
>
> To ensure a fair and rigorous comparison, we applied a 5-fold cross-validation strategy for both datasets. For the BOL dataset, we adapted the initial utterance-based labels by grouping them into rounds, aligning more closely with the video-based labeling format used in the RLT dataset. To address class imbalance within the BOL dataset, we employed random under-sampling of the `deception' class to create a balanced dataset, enabling the model to learn both classes effectively without bias. As a result, the RLT dataset comprised 111 videos, while the balanced BOL dataset included 128 videos. Comprehensive dataset statistics are available in **Appendix A.4**.
>
> The below table in the manuscript presents a performance comparison between AVDDCL and baseline models under these experimental conditions. Notably, our model achieved the **highest performance in the RLT-RLT and BOL-BOL settings, with accuracies of 86.4% and 74%**, respectively. These findings highlight the adaptability and robustness of AVDDCL in addressing diverse deception detection challenges.
>
> By incorporating the RLT and BOL datasets into our analysis, we ensure a more comprehensive evaluation across varied data distributions. The details of these experiments and their outcomes are discussed in **Section 4.5** of the revised manuscript. We once again extend our sincere thanks to the reviewer for their valuable feedback, which helped us address this limitation and expand the scope of our experiments.
>
> This table presents the within-dataset experiments on the RLT (high-stakes) and BOL (low-stakes) datasets. The metrics evaluated are ACC (%), F1-score (%), and AUC (%).
>
>
> | **Model**            | **Modality** | **Train RLT / Test RLT**       |                        |             | **Train BOL / Test BOL**       |                        |             |
> |----------------------|--------------|--------------------------------|------------------------|-------------|--------------------------------|------------------------|-------------|
> |                      |              | **ACC**                       | **F1-score**           | **AUC**     | **ACC**                       | **F1-score**           | **AUC**     |
> | Camara et al. (2024) [1] | V            | 62.7 ± 1.0                         | 62.8 ± 1.0                | 64.1 ± 1.1        | 62.4  ± 1.2                         | 58.2  ± 2.2                 | 62.4  ± 1.2       |
> | Guo et al. (2023) [2]  | V + A        | 82.7 ± 1.3                         | 83.3 ± 1.2                 | 82.8 ± 1.3        | 67.0  ± 1.6                         | 62.1  ± 2.4                 | 67.0  ± 1.2       |
> | **AVDDCL (Ours)**    | V + A        | **86.4 ± 1.4**                      | **85.6 ± 1.6**              | **86.6 ± 1.4**    | **74.0 ± 1.9**                      | **68.5 ± 3.5**              | **74.0 ± 1.9**    |
>
>
>
>
> [1] Camara, M. K., Postal, A., Maul, T. H., & Paetzold, G. H. (2024). Can lies be faked? Comparing low-stakes and high-stakes deception video datasets from a Machine Learning perspective. Expert Systems with Applications, 249, 123684.
>
> [2] Guo, X., Selvaraj, N. M., Yu, Z., Kong, A. W. K., Shen, B., & Kot, A. (2023). Audio-visual deception detection: Dolos dataset and parameter-efficient crossmodal learning. In Proceedings of the IEEE/CVF International Conference on Computer Vision (pp. 22135-22145).

---

> > ### Comment · Reviewer_6r1s · 2024-11-26
> > **Response to Rebuttal**
> >
> > Hello authors!
> > Thank you so much for engaging with my comments. Overall, I am satisfied with the responses. While I would still argue that if imbalance is not so high, then the focal loss is not motivated perfectly; especially L272-276 do not add any new info on why difficult cases might not be possible to 'focus on' with a regular cross-entropy. However overall responses seem to be well addressed! All the best!

---

> ### Author Response · Authors · 2024-11-27
> **Additional Response**
>
> **Question 1: Concern of insufficient justification for focal loss in low-imbalance scenarios, particularly regarding its advantage over cross-entropy for handling difficult cases, is raised in L272-276.**
>
> We sincerely thank the reviewer for their valuable feedback and insightful comments. We agree that the benefits of focal loss may not always be evident when class imbalance is not significant, and we have revised the manuscript to address this.
>
> Regarding L272-276, this section was intended to emphasize the inherent difficulty of detecting deception, which often involves subtle patterns masked by strategic behavior.  While cross-entropy loss treats all samples equally, focal loss introduces a modulating factor that down-weights easy-to-classify samples, prioritizing harder cases like deception.  As described in **Appendix A.3**, this adjustment allows the model to focus on challenging examples, addressing the limitations of cross-entropy loss.
>
> In our experiments, we fixed $\alpha$ to 1 and adjusted $\gamma$ to prioritize difficult samples.  This approach worked effectively on benchmark datasets with relatively small class imbalance.  However, we acknowledge that treating both $\alpha$ and $\gamma$ as learnable parameters could offer greater flexibility and adaptability, particularly in real-time applications or datasets with significant imbalance and complexity.
>
> We have added this discussion to the **Section 5**, highlighting the potential benefits of adaptive parameterization of focal loss for broader applications.
>
> Once again, we deeply appreciate the reviewer’s comments, which have helped us refine our explanations and consider more practical implications for real-world scenarios.

---

### Author Response · Authors · 2024-11-21
**Thanks to all reviewers and PC members**

We would like to express our sincere gratitude to the reviewers and the Program Committee for their valuable time and insightful feedback. We are particularly thankful for your positive recognition of our research direction and for the constructive comments that guided us in improving our work. Moreover, we greatly appreciate the opportunity to revise our paper based on your suggestions, which help enhance the quality of our research.

---

### Note · Authors · 2025-01-23

I have read and agree with the venue's withdrawal policy on behalf of myself and my co-authors.